# MLR-SNet: Transferable LR Schedules for Heterogeneous Tasks

## Abstract

The learning rate (LR) is one of the most important hyper-parameters in stochastic gradient descent (SGD) for deep neural networks (DNN) training and generalization. However, current hand-designed LR schedules need to manually pre-specify a fixed form, which limits their ability to adapt to non-convex optimization problems due to the significant variation of training dynamics. Meanwhile, it always needs to search a proper LR schedule from scratch for new tasks. To address these issues, we propose to parameterize LR schedules with an explicit mapping formulation, called *MLR-SNet*. The learnable structure brings more flexibility for MLR-SNet to learn a proper LR schedule to comply with the training dynamics of DNN. Image and text classification benchmark experiments substantiate the capability of our method for achieving proper LR schedules. Moreover, the meta-learned MLR-SNet is plug-and-play to generalize to new heterogeneous tasks. We transfer our meta-trained MLR-SNet to tasks like different training epochs, network architectures, datasets, especially large scale ImageNet dataset, and achieve comparable performance with hand-designed LR schedules. Finally, MLR-SNet can achieve better robustness when training data are biased with corrupted noise.

## 1 Introduction

Stochastic gradient descent (SGD) and its many variants (Robbins & Monro, 1951; Duchi et al., 2011; Zeiler, 2012; Tieleman & Hinton, 2012; Kingma & Ba, 2015), have been served as the cornerstone of modern machine learning with big data. It has been empirically shown that DNN achieves state-of-the-art generalization performance on a wide variety of tasks when trained with SGD (Zhang et al., 2017). Several recent researches observe that SGD tends to select the so-called flat minima (Hochreiter & Schmidhuber, 1997a; Keskar et al., 2017), which seems to generalize better in practice.

Scheduling learning rate (LR) for SGD is one of the most widely studied aspects to help improve the SGD training for DNN. Specifically, it has been experimentally studied how the LR (Jastrzebski et al., 2017) influences mimima solutions found by SGD. Theoretically, Wu et al. (2018a) analyze that LR plays an important role in minima selection from a dynamical stability perspective. He et al. (2019) provide a PAC-Bayes generalization bound for DNN trained by SGD, which is correlated with LR. In a word, finding a proper LR schedule highly influences the generalization performance of DNN, which has been widely studied recently (Bengio, 2012; Schaul et al., 2013; Nar & Sastry, 2018).

There mainly exist three kinds of hand-designed LR schedules: (1) Pre-defined LR policy is mostly used in current DNN training, like decaying or cyclic LR (Gower et al., 2019; Loshchilov & Hutter, 2017), and brings large improvements in training efficiency. Some theoretical works suggested that the decaying schedule can yield faster convergence (Ge et al., 2019; Davis et al., 2019) or avoid strict saddles (Lee et al., 2019; Panageas et al., 2019) under some mild conditions. (2) LR search methods in tranditional convex optimization (Nocedal & Wright, 2006) can be extended to DNN training by searching LR adaptively in each step, such as Polyak's update rule (Rolinek & Martius, 2018), Frank-Wolfe algorithm (Berrada et al., 2019), and Armijo line-search (Vaswani et al., 2019), etc. (3) Adaptive gradient methods like Adam (Duchi et al., 2011; Tieleman & Hinton, 2012; Kingma & Ba, 2015), adapt LR for each parameters separately according to some gradient information.

Although above LR schedules (as depicted in Fig. 1(a) and 1(b)) can achieve competitive results on their learning tasks, they still have evident deficiencies in practice. On the one hand, these policies need to manually pre-specify the form of LR schedules, suffering from the limited flexibility to adapt to non-convex optimization problems due to the significant variation of training dynamics. On the other hand, when solving new heterogeneous tasks, it always needs to search a proper LR

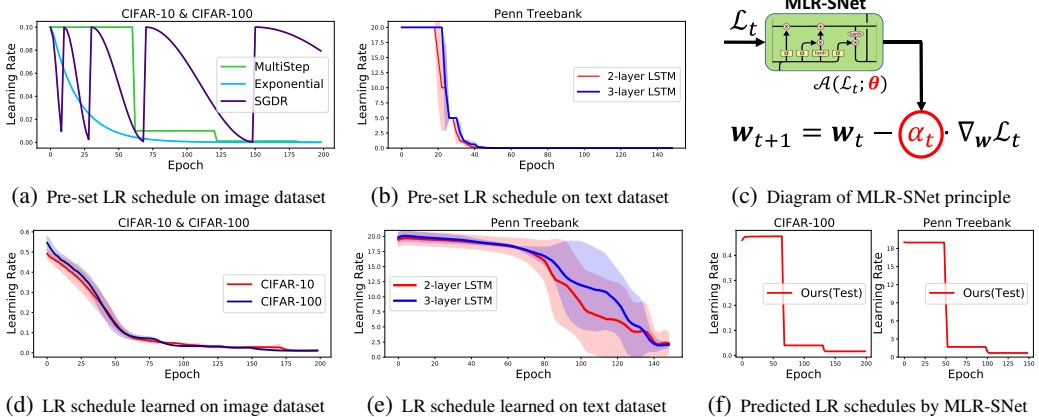

Figure 1: Pre-set LR schedules for (a) image and (b) text classification. (c) Visualization of how we input current loss $\mathcal{L}_t$ to MLR-SNet, which then outputs a proper LR $\alpha_t$ to help SGD find a better minima. LR schedules learned by MLR-SNet on (d) image and (e) text classification. (f) We transfer LR schedules learned on CIFAR-10 to image (CIFAR-100) and text (Penn Treebank) classification, and the subfigure shows the predicted LR during training.

schedule from scratch, as well as to tune their involving hyper-parameters. This process is time and computation expensive, which tends to further raise their application difficulty in real problems.

To alleviate the aforementioned issues, this paper presents a model to learn a plug-and-play LR schedule. The main idea is to parameterize the LR schedule as a LSTM network (Hochreiter & Schmidhuber, 1997b), which is capable of dealing with such a long-term information dependent problem. As shown in Fig. 1(c), the proposed Meta-LR-Schedule-Net (**MLR-SNet**) learns an explicit loss-LR dependent relationship. In a nutshell, this paper makes the following three-fold contributions.

(1) We propose a MLR-SNet to learn an adaptive LR schedule, which can adjust LR based on current training loss as well as the information delivered from past training histories stored in the MLR-SNet. Due to the parameterized form of the MLR-SNet, it can be more flexible than hand-designed policies to find a proper LR schedule for the specific learning task. Fig.1(d) and 1(e) show our learned LR schedules, which have similar tendency as pre-defined policies, but more variations at their locality. This validates the efficacy of our method for adaptively adjusting LR according to training dynamics.

(2) With an explicit parameterized structure, the meta-trained MLR-SNet can be transferred to new heterogeneous tasks (meta-test stage), including different training epochs, network architectures and datasets. Experimental results verify that our plug-and-play LR schedules can achieve comparable performance, while do not have any hyper-parameters compared with tranditional LR schedules. This potentially saves large labor and computation cost in real world applications.

(3) The MLR-SNet is meta-learned to improve generalization performance on unseen data. We validate that with the guidance of clean data, our MLR-SNet can achieve better robustness when training data are biased with corrupted noise than hand-designed LR schedules.

## 2 RELATED WORK

**Meta learning for optimization.** Meta learning has a long history in psychology (Ward, 1937; Lake et al., 2017). Meta learning for optimization can date back to 1980s-1990s (Schmidhuber, 1992; Bengio et al., 1991), aiming to meta-learn the optimization process of learning itself. Recently, Andrychowicz et al. (2016); Ravi & Larochelle (2017); Chen et al. (2017); Wichrowska et al. (2017); Li & Malik (2017); Lv et al. (2017) have attempted to scale this idea to larger DNN optimization problems. The main idea is to construct a meta-learner as the optimizer, which takes the gradients as input and outputs the whole updating rules. These approaches tend to make selecting appropriate training algorithms, scheduling LR and tuning other hyper-parameters in an automatic way. Except for solving continuous optimization problems, some works employ these ideas to other optimization problems, such as black-box functions (Chen et al., 2017), few-shot learning (Li et al., 2017), model's curvature (Park & Oliva, 2019), evolution strategies (Houthooft et al., 2018), combinatorial functions (Rosenfeld et al., 2018), etc.

Though faster in decreasing training loss than the traditional optimizers in some cases, the learned optimizers may not always generalize well to diverse problems, especially longer horizons (Lv et al., 2017) and large scale optimization problems (Wichrowska et al., 2017). Moreover, they can not be

guaranteed to output a proper descent direction in each iteration for DNN training, since they assume all parameters share one small net and ignore the relationship between each parameters. Our proposed method attempts to learn an adaptive LR schedule rather than the whole update rules. This makes it easy to learn and the meta-learned LR schedule can be transferred to new heterogeneous tasks.

**HPO and LR schedule adaptation.** Hyper-parameter optimization (HPO) was historically investigated by selecting proper values for algorithm hyper-parameters to obtain better performance on validation set (see (Hutter et al., 2019) for an overview). Typical methods include grid search, random search (Bergstra & Bengio, 2012), Bayesian optimization (Snoek et al., 2012), gradient-based methods (Franceschi et al., 2017; Shu et al., 2020a;b), etc. Recently, some works attempt to find a proper LR schedule under the framework of gradient-based HPO, which can be solved by bilevel optimization (Franceschi et al., 2017; Baydin et al., 2018). However, most HPO techniques tend to fall into short-horizon bias and easily find a bad minima (Wu et al., 2018b). Our MLR-SNet has an explicit function form, which makes the optimization of the LR schedules more robust and effective.

**Transfer to heterogeneous tasks.** Transfer learning (Pan & Yang, 2009) aims to transfer knowledge obtained from source task to help the learning on the target task. Most transfer learning methods assume the source and target tasks consist of the same instance, feature or model spaces (Yang et al., 2020), which greatly limits their applications. Recently, meta learning (Finn et al., 2017) aims to learn common knowledge shared over a distribution of tasks, such that the learned knowledge can transfer to unseen heterogeneous tasks. Most meta learning approaches focus on few shot learning framework, while we attempt to extend it into a standard learning framwork. The hand-designed LR schedules and HPO methods just try to find a proper LR schedule for given tasks, and need to be learned from scratch for new tasks. However, our meta-learned MLR-SNet is plug-and-play, which can directly transfer how to schedule LR for SGD to heterogeneous tasks without additional learning.

## 3 THE PROPOSED META-LR-SCHEDULE-NET (MLR-SNET) METHOD

The problem of training DNN can be formulated as the following non-convex optimization problem,

$$\min_{w \in \mathbb{R}^n} \mathcal{L}_{Tr}(D_{Tr}; w) := \frac{1}{N} \sum_{i=1}^{N} \mathcal{L}_i^{Tr}(w), \tag{1}$$

where $\mathcal{L}_i^{Tr}$ is the training loss function for data samples $i \in D_{Tr} = \{1, 2, \cdots, N\}$, which characters the deviation of the model prediction from the data, and $w \in \mathbb{R}^n$ represents the parameters of the model (e.g., the weight matrices in DNN) to be optimized. SGD (Robbins & Monro, 1951; Polyak, 1964) and its variants, including Momentum (Tseng, 1998), Adagrad (Duchi et al., 2011), Adadelta (Zeiler, 2012), RMSprop (Tieleman & Hinton, 2012), Adam (Kingma & Ba, 2015), are often used for training DNN. In general, these algorithms can be summarized as the following formulation,

$$w_{t+1} = w_t + \Delta w_t, \Delta w_t = \mathcal{O}_t(\nabla \mathcal{L}^{Tr}(w_t), \mathcal{H}_t; \Theta_t), \tag{2}$$

where $w_t$ is $t$-th updating model parameters, $\nabla \mathcal{L}^{Tr}(w_t)$ denotes the gradient of $\mathcal{L}^{Tr}$ at $w_t$, $\mathcal{H}_t$ represents the historical gradient information, and $\Theta_t$ is the hyperparameter of the optimizer $\mathcal{O}$, e.g., LR. To present our method's efficiency, we focus on the following vanilla SGD formulation,

$$w_{t+1} = w_t - \alpha_t \left( \frac{1}{|B_t|} \sum_{i \in B_t} \nabla \mathcal{L}_i^{Tr}(w_t) \right), \tag{3}$$

where $B_t \subset D_{Tr}$ denotes the batch samples randomly sampled from the training dataset, $|B_t|$ denotes the number of the sampled batch samples, and $\nabla \mathcal{L}_i^{Tr}(w_t)$ denotes the gradient of sample $i$ computed at $w_t$ and $\alpha_t$ is the LR at $t$-th iteration.

### 3.1 EXISTING LR SCHEDULE STRATEGIES

As Bengio (2012) demonstrated, the choice of LR remains central to effective DNN training with SGD. As mentioned in Section 1, a variety of hand-designed LR schedules have been proposed. Though they achieve competitive results on some learning tasks, they share several drawbacks: (1) The pre-defined LR schedules suffer from the limited flexibility to adapt to the significantly changed training dynamics for the non-convex optimization problems. (2) It needs to be learned from scratch to find a proper LR schedule for the new tasks, which raises their application difficulty in real problems.

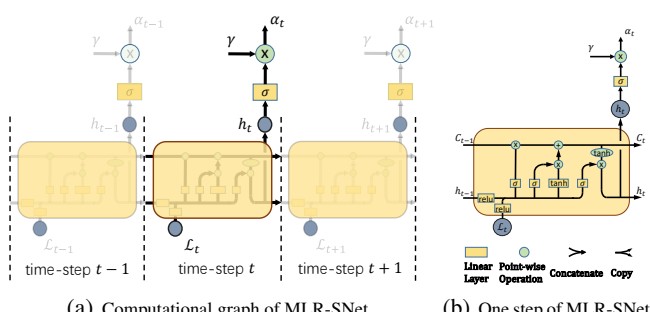

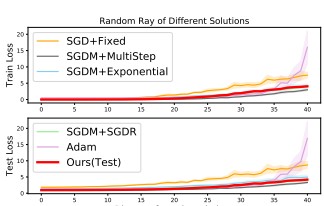

(a) Computational graph of MLR-SNet

(b) One step of MLR-SNet

Figure 2: The structure of our proposed MLR-SNet.

Figure 3: (**Above**) Train loss and (**Below**) test loss as a function of a point on a random ray starting at the solutions for different methods on CIFAR-100 with ResNet-18.

Inspired by current meta-learning developments (Finn et al., 2017; Shu et al., 2018; 2019), some researches proposed to learn a generic optimizer from data (Andrychowicz et al., 2016; Ravi & Larochelle, 2017; Chen et al., 2017; Wichrowska et al., 2017; Li & Malik, 2017; Lv et al., 2017). The main idea is to learn a meta-learner as the optimizer to guide the learning of the whole updating rules. For example, Andrychowicz et al. (2016) try to replace Eq.(2) with the following formulation,

$$w_{t+1} = w_t + g_t, [g_t, h_{t+1}]^T = m(\nabla_t, h_t; \phi), \tag{4}$$

where $g_t$ is the output of a LSTM net $m$, parameterized by $\phi$, whose state is $h_t$. This strategy can make selecting appropriate training algorithms, scheduling LR and tuning other hyper-parameters in a unified and automatic way. Though faster in decreasing training loss than the traditional optimizers in some cases, the learned optimizer may not always generalize well to more variant and diverse problems, like longer horizons (Lv et al., 2017) and large scale optimization problems (Wichrowska et al., 2017). Moreover, it can not guarantee to output a proper descent direction in each iteration for network training. This tends to further increase their application difficulty in real problems.

Recently, some methods (Franceschi et al., 2017; Baydin et al., 2018) consider the following constrained optimization problem to search the optimal LR schedule $\alpha^*$ such that the produced models are associated with small validation error,

$$\min_{\alpha=\{\alpha_0,\cdots,\alpha_{T-1}\}} \mathcal{L}_{Val}(D_{Val}, w_T), \ s.t. \ w_{t+1} = \phi_t(w_t, \alpha_t), \ t = 0, 1, \cdots, T-1, \tag{5}$$

where $\mathcal{L}_{Val}$ denotes the validation loss function, $D_{Val} = \{1, 2, \cdots, M\}$ denotes hold-out validation set, $\alpha$ is to-be-solved hyper-parameter, $\phi_t : \mathbb{R}^n \times \mathbb{R}_+ \to \mathbb{R}^n$ is a stochastic weight update dynamics, like the updating rule in Eq.(2) or the vanilla SGD in Eq.(3), and $T$ is the maximum iteration step. Though achieving comparable results on some tasks with hand-designed LR schedules, they can not directly transfer to new tasks, since they do not have an explict transferable structure form.

### 3.2 PROPOSED META-LR-SCHEDULE-NET (MLR-SNET) METHOD

To address aforementioned issues, the main idea is to design a meta-learner with an explicit mapping formulation to parameterize LR schedules as shown in Fig.1(c), called MLR-SNet. The parameterized structure can bring two benefits: 1) It gives more flexibility to learn a proper LR schedule to comply with the significantly changed training dynamics of DNN; 2) It makes the meta-learned LR schedules be transferable and plug-and-play, which can be applied to new heterogeneous tasks.

**Formulation of MLR-SNet.** The computational graph of MLR-SNet is depicted in Fig.2(a). Let $\mathcal{A}(\cdot; \theta)$ denote the MLR-SNet, and then the updating equation of SGD in Eq.(3) can be rewritten as

$$w_{t+1} = w_t - \mathcal{A}(\mathcal{L}_t; \theta_t) \left( \frac{1}{|B_t|} \sum_{i \in B_t} \nabla \mathcal{L}_i^{Tr}(w_t) \right), \mathcal{L}_t = \frac{1}{|B_t|} \sum_{i \in B_t} \mathcal{L}_i^{Tr}(w_t), \tag{6}$$

where $\theta_t$ is the parameter of MLR-SNet at $t$-th iteration ($t = 0, \cdots, T-1$). At any iteration steps, $\mathcal{A}(\cdot; \theta)$ can learn an explicit loss-LR dependent relationship, such that the net can adaptively predict LR according to the current input loss $\mathcal{L}_t$, as well as the historical information stored in the net. For every iteration step, the whole forward computation process is (as shown in Fig. 2(b))

$$\begin{pmatrix} i_t \\ f_t \\ o_t \\ g_t \end{pmatrix} = \begin{pmatrix} \sigma \\ \sigma \\ \sigma \\ \tanh \end{pmatrix} W_2 \begin{pmatrix} \text{ReLU} \\ \text{ReLU} \end{pmatrix} W_1 \begin{pmatrix} h_{t-1} \\ \mathcal{L}_t \end{pmatrix}, \begin{matrix} c_t = f_t \odot c_{t-1} + i_t \odot g_t \\ h_t = o_t \odot \tanh(c_t) \\ p_t = \sigma(W_3 h_t) \\ \alpha_t = \gamma \cdot p_t \end{matrix}, \tag{7}$$

where $i_t, f_t, o_t$ denote the Input, Forget and Output gates, respectively. Different from vanilla LSTM, the input $h_{t-1}$ and the training loss $\mathcal{L}_t$ are preprocessed by a fully-connected layer $W_1$ with ReLU activation function. Then it works as LSTM and obtains the output $h_t$. After that, the predicted value $p_t$ is obtained by a linear transform $W_3$ on the $h_t$ with a Sigmoid activation function. Finally, we introduce a scale factor $\gamma$ [1] to guarantee the final predicted LR located in the interval of $[0, \gamma]$. Albeit simple, this net is known for dealing with such long-term information dependent problem, and thus capable of finding a proper LR schedule to comply with the complex variations of training dynamics.

**Meta-Train: adapting to the training dynamics of DNN.** The MLR-SNet can be meta-trained to improve the generalization performance on unseen validation data for DNN training as follows:

$$\min_\theta \mathcal{L}_{Val}(D_{Val}, w_T), \ s.t. \ w_{t+1} = \phi_t(w_t, \mathcal{A}(\mathcal{L}_t; \theta)), \ t = 0, 1, \cdots, T-1. \tag{8}$$

Now the important question is how to efficiently meta-learn the parameter $\theta$ for the MLR-SNet. We employ the online approximation technique in (Shu et al., 2019) to jointly update $\theta$ and model parameter $w$ to explore a proper LR schedule with better generalization for DNNs training. However, the step-wise optimization for $\theta$ is still expensive to handle large-scale datasets and DNN. Furthermore, we attempt to update $\theta$ after updating $w$ several steps ($T_{val}$) as summarized in Algorithm 1.

Updating $\theta$. When it does not satisfy the updating conditions, $\theta$ is fixed; otherwise, $\theta$ is updated using the model parameter $w_t$ and MLR-SNet parameter $\theta_t$ obtained in the last step by minimizing the validation loss defined in Eq.(8). Adam can be employed to optimize the validation loss, i.e.,

$$\theta_{t+1} = \theta_t + Adam(\nabla_\theta \mathcal{L}_{Val}(D_m, \hat{w}_{t+1}(\theta)); \eta_t), \tag{9}$$

where $Adam$ denotes the Adam algorithm, whose input is the gradient of validation loss with respect to MLR-SNet parameter $\theta$ on $m$ mini-batch samples $D_m$ from $D_{Val}$. $\eta_t$ denotes the LR of Adam. $\hat{w}_{t+1}(\theta)$ [2] is formulated on a mini-batch training samples $D_n$ from $D_{Tr}$ as follows:

$$\hat{w}_{t+1}(\theta) = w_t - \mathcal{A}(\mathcal{L}_{Tr}(D_n, w_t); \theta) \cdot \nabla_w \mathcal{L}_{Tr}(D_n, w)\big|_{w_t}. \tag{10}$$

Updating $w$. Then, the updated $\theta_{t+1}$ is employed to ameliorate the model parameter $w$, i.e.,

$$w_{t+1} = w_t - \mathcal{A}(\mathcal{L}_{Tr}(D_n, w_t); \theta_{t+1}) \cdot \nabla_w \mathcal{L}_{Tr}(D_n, w)\big|_{w_t}. \tag{11}$$

The whole meta-train learning algorithm can be summarized in Algorithm 1. All computations of gradients can be efficiently implemented by automatic differentiation libraries, like PyTorch (Paszke et al., 2019), and generalized to any DNN architectures. It can be seen that the MLR-SNet can be gradually optimized during the learning process and adjust the LR dynamically based on the training dynamics of DNNs.

---

**Algorithm 1** The Meta-Train Algorithm of MLR-SNet

**Input:** Training data $D_{Tr}$, validation set $D_{Val}$, batch size $n, m$, max iterations $T$, updating period $T_{val}$.
**Output:** Model parameter $w_T$ and MLR-SNet parameter $\theta_T$
1: Initialize model parameter $w_0$ and MLR-SNet parameter $\theta_0$.
2: **for** $t = 0$ **to** $T - 1$ **do**
3:     $D_n \leftarrow$ SampleMiniBatch($D_{Tr}, n$).
4:     **if** $t \% T_{val} = 0$, **then**
5:        $D_m \leftarrow$ SampleMiniBatch($D_{Val}, m$).
6:        Update $\theta_{t+1}$ by Eq. (9).
7:     **end if**
8:     Update $w_{t+1}$ by Eq. (11).
9: **end for**

---

**Meta-Test: transferring to heterogeneous tasks.** When we obtain the meta-learned MLR-SNet, it can be easily applied to new tasks. Now the new model parameter $u$ for the new task is updated by,

$$u_{t+1} = u_t - \mathcal{A}(\mathcal{L}_{Tr}(D_n, u_t); \theta^*) \cdot \nabla_u \mathcal{L}_{Tr}(D_n, u)\big|_{u_t}, \tag{12}$$

where $\theta^*$ is the parameter of the meta-learned MLR-SNet, which is fixed in the meta-test stage.

## 4 EXPERIMENTAL RESULTS

To evaluate the proposed MLR-SNet, we firstly conduct experiments to show our method is capable of finding proper LR schedules compared with baseline methods. Then we transfer the learned LR schedules to various tasks to show its superiority in generalization. Finally, we show our method behaves robust and stable when training data contain different data corruptions.

---

[1] As we know that the performance of hand-designed LR schedules and HPO methods is very sensitive to the initial LR. To avoid carefully tuning the initial LR, we learn the LR schedules from an interval $[0, \gamma]$, and now the initial LR is determined by the output of the MLR-SNet. We set $\gamma = 1$ for image tasks, and $\gamma = 40$ for text tasks in all our experiments to eliminate the influence of loss magnitude between two different tasks.

[2] Notice that $\hat{w}_{t+1}(\theta)$ here is a function of $\theta$ to guarantee the gradient in Eq.(9) to be able to compute.

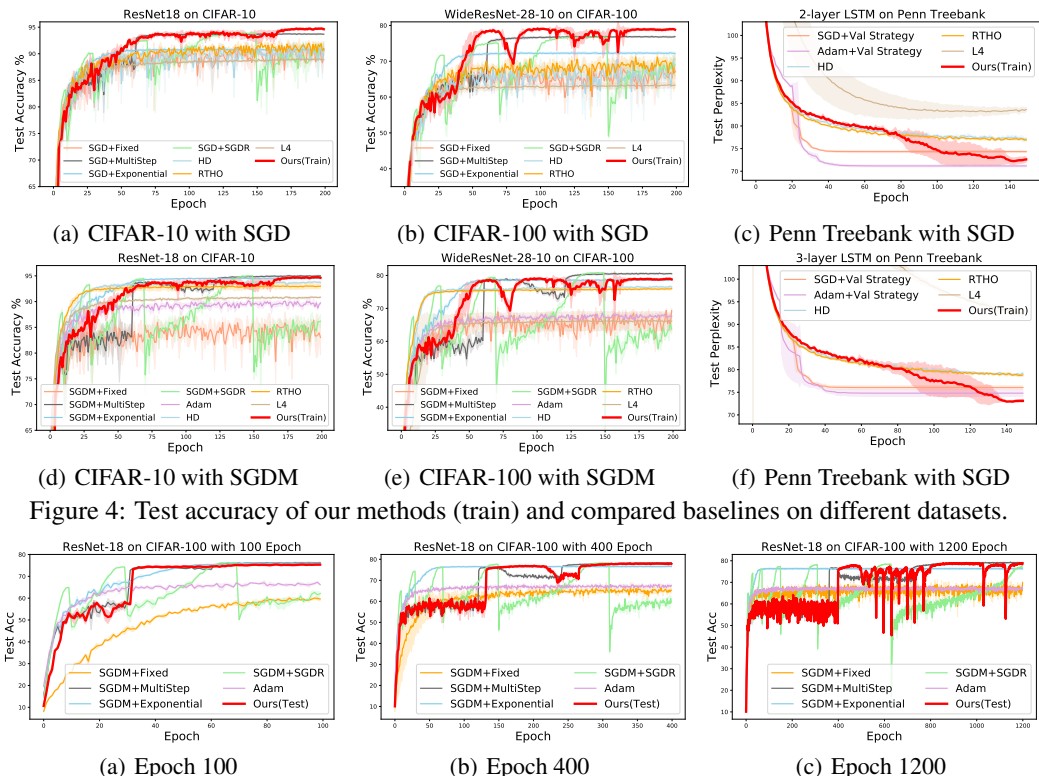

Figure 4: Test accuracy of our methods (train) and compared baselines on different datasets.

Figure 5: Test accuracy on CIFAR-100 of ResNet-18 with varying epochs.

## 4.1 META-TRAIN: EVALUATION ON THE LR SCHEDULE LEARNED BY MLR-SNET

**Datasets and models.** To verify general effectiveness of our method, we respectively train different models on four benchmark data, including ResNet-18 (He et al., 2016) on CIFAR-10, WideResNet-28-10 (Zagoruyko & Komodakis, 2016) on CIFAR-100 (Krizhevsky, 2009), 2-layer LSTM and 3-layer LSTM on Penn Treebank (Marcus & Marcinkiewicz).

**Baselines.** For image classification tasks, the compared methods include SGD with hand-designed LR schedules: 1) **Fixed** LR, 2) **Exponential** decay, 3) **MultiStep** decay, 4) SGD with restarts (**SGDR**) (Loshchilov & Hutter, 2017). Also, we compare with SGD with Momentum (SGDM) with above four LR schedules. The momentum is fixed as 0.9. Meanwhile, we compare with adaptive gradient method: 5)**Adam**, LR search method: 6) **L4** (Rolinek & Martius, 2018), and current LR schedule adaptation methods: 7) hyper-gradient descent (**HD**) (Baydin et al., 2018), 8) real-time hyper-parameter optimization (**RTHO**) (Franceschi et al., 2017). For text classification tasks, we compare with 1) SGD and 2) Adam with LR tuned using a validation set. They drop the LR by a factor of 4 when the validation loss stops decreasing. Also, we compared with 3) **L4**, 4) **HD**, 5) **RTHO**. We run all experiments with 3 different seeds reporting accuracy. The detailed illustrations of experimental setting, and more experimental results are presented in Appendix B.

**Image tasks.** Fig.4(a) and 4(b) show the classification accuracy on CIFAR-10 and CIFAR-100 test sets, respectively. It can be observed that: 1) our algorithm outperforms all other competing methods, and the learned LR schedules by MLR-SNet are presented in Fig.1(d), which have similar shape as the hand-designed policies, while with more elaborate variation details in locality for adapting training dynamics. 2) The Fixed LR has similar performance to other baselines at the early training, while falls into fluctuations at the later training. This implies that the Fixed LR can not finely adapt to such DNN training dynamics. 3) The MultiStep LR drops the LR at some epochs, and such elegant strategy overcomes the issue of Fixed LR and obtains higher and stabler performance at the later training. 3) The Exponential LR improves test performance faster at the early training than other baselines, while makes a slow progress due to smaller LR at the later training. 4) SGDR uses the cyclic LR, which needs more epochs to obtain a stable result. 5) Though Adam has an adaptive coordinate-specific LR, it behaves worse than MultiStep and Exponential LR as demonstrated in Wilson et al. (2017). An extra tuning is necessary for better performance. 6) L4 greedily searches LR locally to decrease loss, while the complex DNN training dynamics can not guarantee it to obtain a good minima. 7) HD and RTHO are able to achieve similar performance to hand-designed LR schedules. The LR schedules learned by L4, HD and RTHO can be found in supplementary material.

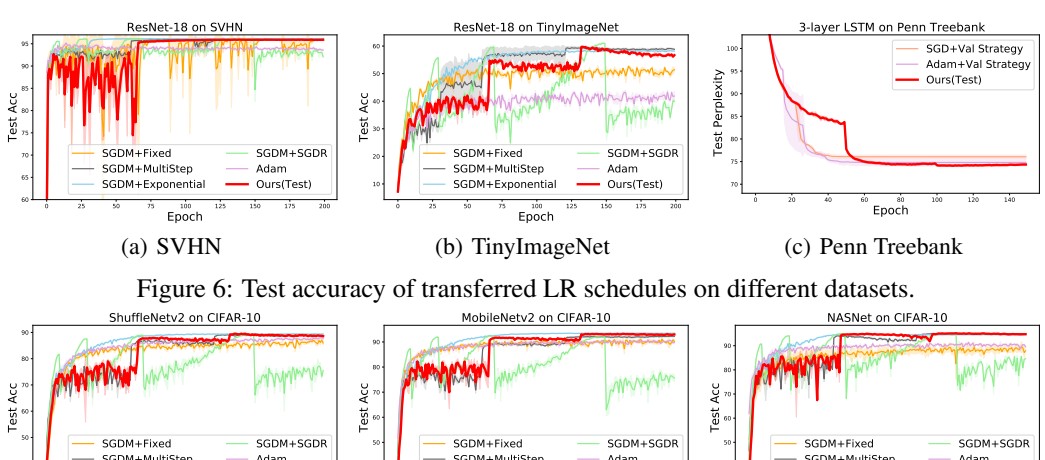

Figure 6: Test accuracy of transferred LR schedules on different datasets.

Figure 7: Test accuracy on CIFAR-10 of different network architectures

Since image tasks often use SGDM to train DNNs, Fig.4(d) and 4(e) show the results of baseline methods trained with SGDM, and they obtain a remarkable improvement than SGD. Though not using extra historical gradient information to help optimization, our method achieves comparable results with baselines by finding a proper LR schedule for SGD.

**Text tasks.** Fig.4(c) and 4(f) show the test perplexity on the Penn Treebank with 2-layer and 3-layer LSTM, respectively. Adam and SGD heuristically drops LR when the validation loss stops decreasing. However, our MLR-SNet predicts LR according to training dynamics by minimizing the validation loss, which is a more intelligent way to employ the validation dataset. Thus our method achieves comparable or even better performance than Adam and SGD. The learned LR schedules of the MLR-SNet are presented in Fig.1(b), which have similar shape as the hand-designed policies. L4 often falls into a bad minima since it greedily searches LR locally. HD and RTHO directly optimize LR to improve the performance on validation dataset, obtaining the similar results as Adam and SGD. With an explicit strcuture, our method behaves more robust and efficient than HD and RTHO.

**Remark.** Actually, the performance of the hand-design LR schedules can be regarded as the best/ upper performance bound. Since these strategies have been tested to work well for the specific tasks, and they are written into the standard deep learning library. For different image and text tasks, our MLR-SNet can achieve the similar or even a little better performance compared with the best baselines, demostrating the effectiveness and generality of our method.

### 4.2 META-TEST: TRANSFERABILITY OF PLUG-AND-PLAY LR SCHEDULES

The learned MLR-SNet is transferable and plug-and-play. Here we validate if MLR-SNet can transfer to new heterogeneous tasks. Since the methods L4,HD,RTHO in Section 4.1 are not able to generalize, we do not compare them here. Actually our results show superiority on image tasks beyond baseline methods when trained with SGD, here we present stronger baseline results in which compared methods are trained with SGDM. We use the MLR-SNet meta-learned on CIFAR-10 with ResNet-18 in Section 4.1 as the plug-and-play LR schedules for the following experiments.

**Transfer to different epochs.** The plug-and-play MLR-SNet is meta-trained with epoch 200, and we transfer it to other different training epochs, e.g., 100, 400,1200. As shown in Fig.5, our MLR-SNet has the ability to train for longer horizons and achieves almost same performance as MultiStep LR. The slight shakes for epoch 1200 may due to that our MLR-SNet can learn the LR similar to SGDR locally. The Exponential LR has a little performance decreased for the longer epochs.

**Transfer to different datasets.** We transfer the LR schedules meta-learned on CIFAR-10 to SVHN (Netzer et al., 2011), TinyImageNet [3], and Penn Treebank (Marcus & Marcinkiewicz). As shown in Fig.6, though datasets vary from image to text, our method can still obtain a relatively stable and comparable generalization performance for different tasks with baseline method.

**Transfer to different net architectures.** We also transfer the LR schedules meta-learned on ResNet-18 to light-weight nets ShuffleNetV2 (Ma et al., 2018), MobileNetV2 (Sandler et al., 2018) or NASNet

---

[3]It can be downloaded at https://tiny-imagenet.herokuapp.com.

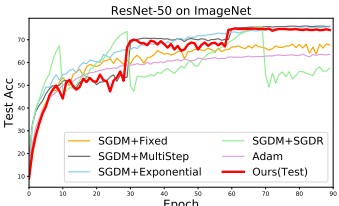
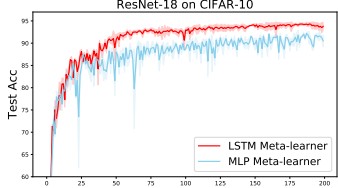
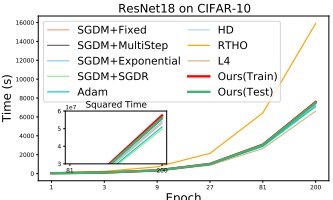

Figure 8: Test accuracy on Ima-geNet with ResNet-50.

Figure 9: Comparion of different meta-learners.

Figure 10: Time consuming of different LR schedules methods.

Table 1: Test accuracy (%) on CIFAR-10 and CIFAR-100 training set of different methods trained on CIFAR-10-C and CIFAR-100-C. Best and Last denote the results of the best and the last epoch.

| Datasets/Methods | | Fixed | MultiStep | Exponential | SGDR | Adam | Ours(Train) |
|---|---|---|---|---|---|---|---|
| CIFAR-10-C | Best | 79.78±3.95 | 85.52±1.72 | 83.48±1.45 | 85.94±1.52 | 81.45±1.42 | **86.04±1.51** |
| | Last | 77.88±3.91 | 85.36±1.71 | 83.32±1.43 | 78.21±2.01 | 80.29±1.64 | **85.87±1.54** |
| CIFAR-100-C | Best | 46.74±3.03 | 52.26±2.58 | 49.72±1.97 | 52.54±2.49 | 45.45±1.94 | **52.56±2.26** |
| | Last | 44.79±3.91 | 52.16±2.59 | 49.58±1.98 | 41.58±3.24 | 43.76±2.22 | **52.42±2.34** |

(Zoph et al., 2018) [4]. As shown in Fig.7, our method achieves almost similar results to SGDM with MultiStep or Exponential LR.

**Transfer to large scale optimization problem.** To our best knowledge, only Wichrowska et al. (2017) attempted to use the learned optimizers to train DNN on ImageNet dataset (Deng et al., 2009) among existing learning-to-optimize literatures. However, it can only be executed for thousands of steps, and then its loss begins to increase dramatically, far from the optimization process in practice. We transfer the LR schedule meta-trained on CIFAR-10 with ResNet-18 to ImageNet dataset with ResNet-50 [5]. As shown in Fig.8, the validation accuracy of our method is competitive with those hand-designed LR schedules methods. This implies our method is capable of dealing with such large scale optimization problem, making learning-to-optimize ideas towards more practical applications.

### 4.3 ROBUSTNESS ON DIFFERENT DATA CORRUPTIONS

In this section, we validate that whether our MLR-SNet behaves robust against corrupted training data. To this aim, we design experiments as follows: we take CIFAR-10-C and CIFAR-100-C (Hendrycks & Dietterich, 2019) as our training set, consisting of 15 types of different generated corruptions on test images data of CIFAR-10/CIFAR-100, and the original training set of CIFAR-10/100 as test set. Though the original images of CIFAR-10/100-C are the same with the CIFAR-10/100 test set, different corruptions have changed the data distributions. To guarantee the calculated models finely generalize to test set, we choose the validation set as 10 clean images for each class. Each corruption can be roughly regarded as a task, and thus we obtain 15 models trained on CIFAR-10/100-C. Table 1 shows the mean test accuracy of 15 models (±std). As can be seen, our proposed MLR-SNet is capable of achieving better generalization performance on clean test data than baseline methods, which implies that our method behaves more robust and stable than the pre-set LR schedules when the learning tasks are changed. This is due to that our MLR-SNet has more flexibility to adapt the variation of the data distribution than the pre-set LR schedules, and it can find a proper LR schedule through minimizing the generalization error which is based on the knowledge specifically conveyed from the given validation data. The detailed illustrations of experimental setting, and the transferrablity experiment of meta-learned MLR-SNet are presented in supplementary material.

## 5 SOME ANALYSIS OF MLR-SNET

### 5.1 CONVERGENCE ANALYSIS OF MLR-SNET

The preliminary experimental evaluations show that our method gives good convergence performance on various tasks. We find that the meta-learned LR schedules in our experiments follow a consistent trajectory as shown in Fig.1, sharing a similar tendency as the Exponential LR schedules. To provide a theoretical convergence analysis, we roughly assume that the LR predicted by MLR-SNet obey a Exponential LR form. The convergence analysis for DNN training can refer to (Li et al., 2020). Here, we provide a convergence analysis of the MLR-SNet training. The proof is listed in the Appendix A.

**Theorem 1** *Suppose the loss function $\ell$ is Lipschitz smooth with respect to the model parameter $w$ with constant $L$, and have $\rho$-bounded gradients with respect to training/validation data. And the $\mathcal{A}(\theta)$ is differential with a $\delta$-bounded gradient and twice differential with its Hessian bounded*

---

[4]The pytorch code of all these networks can be found on https://github.com/weiaicunzai/pytorch-cifar100.

[5]The training code can be found on https://github.com/pytorch/examples/tree/master/imagenet.

*by $\mathcal{B}$. Let the learning rate $\alpha_t = \mathcal{A}(\theta_t)$ predicted by MLR-SNet obey the exponential LR, i.e., $\alpha_t = \alpha_0 \beta^t, \beta = (\Gamma/T)^{1/T}, \Gamma \geq 1$. Let $\eta_t = \eta$ for all $t \in [T]$. If we use Adam algorithm to update MLR-SNet, we choose $\eta$ satisfied $\eta \leq \frac{\epsilon}{2L}$ and $1 - \beta_2 \leq \frac{\epsilon^2}{16\rho^2}$, where $\beta_2, \epsilon$ are the hyperparameter of the Adam algorithm. Then for $\theta_t$ generated using Adam, we have the following bound:*

$$\min_{0 \leq t \leq T} \mathbb{E}[\|\nabla \mathcal{L}_{Val}(\hat{\mathbf{w}}_t(\theta_t))\|_2^2] \leq \mathcal{O}(\frac{C \ln(T)}{T} + \sigma^2), \tag{13}$$

*where $C$ is some constant independent of the convergence process, $\sigma$ is the variance of drawing uniformly mini-batch sample at random.*

## 5.2 THE STRUCTURE OF THE MLR-SNET

We regard the LR scheduling as a long-term information dependent problem, and thus we parameterize the LR schedule as an LSTM network. As we known, MLP network can also learn an explicit mapping but ignore the temporal information. Here, we compare the performance of the two types of meta-learners. As shown in Fig. 9, in the early training stage, both of them achieve the similar performance. While at the later training stage, the LSTM meta-learner brings a notable performance increase compared with MLP meta-learner. This may due to that the accumulated temporal information of the LSTM meta-learner can help find a more proper LR for such DNNs training.

## 5.3 COMPUTATIONAL COMPLEXITY ANALYSIS

In the meta-training stage, our MLR-SNet learning algorithm can be roughly regarded as requiring two extra full forward and backward passes of the network (step 6 in algorithm 1) in the presence of the normal network parameters update (step 8 in algorithm 1), together with the forward passes of MLR-SNet for every LR. Therefore compared to normal training, our method needs about $3\times$ computation time for one iteration. Since we periodically update MLR-SNet after several iterations, this will not substantially increase the computational complexity compared with normal network training. In the meta-test stage, our transferred LR schedules predict LR for each iteration by a small MLR-SNet, whose computational cost should be significantly less than the cost of the normal network training. To empirically show the differences between hand-designed LR schedules and our method, we conduct experiments with ResNet-18 on CIFAR-10 and report the running time for all methods. All experiments are implemented on a computer with Intel Xeon(R) CPU E5-2686 v4 and a NVIDIA GeForce RTX 2080 8GB GPU. We follow the corresponding settings in Section 4.1, and results are shown in Figure 10. Except that **RTHO** costs significantly more time, other methods including MLR-SNet training and testing have similar time consuming. Our MLR-SNet takes barely longer time to complete the meta-training and meta-testing phase compared to hand-designed LR schedules. Therefore our method is completely capable of practical application.

## 6 CONCLUSION AND DISCUSSION

In this paper, we have proposed to learn an adaptive and transferrable LR schedule in a meta learning manner. To this aim, we design an LSTM-type meta-learner (MLR-SNet) to parameterize LR schedules, which gives more flexibility to adaptively learn a proper LR schedule to comply with the significantly complex training dynamics of DNN. Meanwhile, the meta-learned LR schedules are plug-and-play and transferrable, which can be transferred how to schedule LR for SGD to new heterogeneous tasks. Comprehensive experiments substantiate the superiority of our method on various image and text benchmarks in its adaptability, transferrability and robustness, as compared with current LR schedules policies. The MLR-SNet is highly practical as it requires negligible increase in the parameter size and computation time, and no transferrable cost for new tasks. We believe our proposed method has a potential to become a new tool to study how to design LR schedules to help improve current DNN training, as well as more practical applications.

Recently, Keskar et al. (2017); Dinh et al. (2017) suggested that the width of a local optimum is related to generalization. Wider optima leads to better generalization. We use the visualization technique in (Izmailov et al., 2018) to visualize the "width" of the solutions for different LR schedules on CIFAR-100 with ResNet-18. As shown in Fig.3, our method lies a wide flat region of the train loss. This could explain the better generalization of our method compared with pre-set LR schedules. Deeper understandings on this point will be further investigated.

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

# A CONVERGENCE ANALYSIS OF THE MLR-SNET

**Lemma 1** *Suppose the loss function $\ell$ is Lipschitz smooth with respect to the model parameter $w$ with constant $L$, and have $\rho$-bounded gradients with respect to training/validation data. And the $\mathcal{A}(\theta)$ is differential with a $\delta$-bounded gradient and twice differential with its Hessian bounded by $\mathcal{B}$. Then the gradient of MLR-SNet parameter $\theta$ with respect to loss is Lipschitz smooth.*

**Proof** *The gradient of MLR-SNet parameter $\theta$ with respect to loss*

$$\nabla_\theta \ell_j(\hat{w}_t(\theta))|_{\theta_t} = \frac{\partial \ell_j(\hat{w}_t(\theta))}{\partial \hat{w}_t(\theta)} \frac{\partial \hat{w}_t(\theta)}{\partial \mathcal{A}(\theta)} \frac{\partial \mathcal{A}(\theta)}{\partial \theta}$$

$$= \frac{-\alpha_t}{n} \sum_{i=1}^{n} \left( \frac{\partial \ell_j(\hat{w}_t(\theta))}{\partial \hat{w}_t(\theta)} \frac{\partial \ell_i(w_t)}{\partial w_t} \right) \frac{\partial \mathcal{A}(\theta)}{\partial \theta}|_{\theta_t},$$

*Let $G_{ij} = \frac{\partial \ell_j(\hat{w}_t(\theta))}{\partial \hat{w}_t(\theta)} \frac{\partial \ell_i(w_t)}{\partial w_t}$, and then take gradient of $\theta$ in both sides of above equallity, we have*

$$\nabla_{\theta^2}^2 \ell_j(\hat{w}_t(\theta))|_{\theta_t} = \frac{-\alpha_t}{n} \sum_{i=1}^{n} \left[ \frac{\partial G_{ij}}{\partial \theta} \frac{\partial \mathcal{A}(\theta)}{\partial \theta} + G_{ij} \frac{\partial \mathcal{A}^2(\theta)}{\partial \theta^2} \right]. \tag{14}$$

*For the first term in the right hand side, we have that*

$$\left\| \frac{\partial G_{ij}}{\partial \theta} \frac{\partial \mathcal{A}(\theta)}{\partial \theta} \right\| \leq \delta \left\| \frac{\partial \ell_j(\hat{w}_t(\theta))}{\partial \hat{w}_t(\theta) \partial \theta} \frac{\partial \ell_i(w_t)}{\partial w_t} \right\|$$

$$= \delta \left\| \frac{\partial}{\partial \hat{w}_t(\theta)} \left( \frac{-\alpha_t}{n} \sum_{i=1}^{n} \left( \frac{\partial \ell_j(\hat{w}_t(\theta))}{\partial \hat{w}_t(\theta)} \frac{\partial \ell_i(w_t)}{\partial w_t} \right) \frac{\partial \mathcal{A}(\theta)}{\partial \theta}|_{\theta_t} \right) \frac{\partial \ell_i(w_t)}{\partial w_t} \right\|$$

$$= \delta \left\| \left( \frac{-\alpha_t}{n} \sum_{i=1}^{n} \left( \frac{\partial^2 \ell_j(\hat{w}_t(\theta))}{\partial \hat{w}_t^2(\theta)} \frac{\partial \ell_i(w_t)}{\partial w_t} \right) \frac{\partial \mathcal{A}(\theta)}{\partial \theta}|_{\theta_t} \right) \frac{\partial \ell_i(w_t)}{\partial w_t} \right\| \tag{15}$$

$$\leq \alpha_t L \rho^2 \delta^2.$$

*For the second term in the right hand side, we have that*

$$\left\| G_{ij} \frac{\partial \mathcal{A}^2(\theta)}{\partial \theta^2} \right\| \leq \mathcal{B} \rho^2 \tag{16}$$

*Combining the above two inequalities Eq.(15)(16), we have*

$$\|\nabla_\theta \ell_j(\hat{w}_t(\theta))|_{\theta_t}\| \leq \alpha \rho^2 (\alpha_t L \delta^2 + \mathcal{B}). \tag{17}$$

*Define $L_A = \alpha \rho^2 (\alpha_t L \delta^2 + \mathcal{B})$, and based on the Lagrange mean value theorem, we have:*

$$\|\nabla \mathcal{L}_{Val}(\hat{\mathbf{w}}_t(\theta_1)) - \mathcal{L}_{Val}(\hat{\mathbf{w}}_t(\theta_2))\| \leq L_A \|\theta_1 - \theta_2\|. \tag{18}$$

*Thus the conclusion holds.*

**Theorem 2** *Suppose the loss function $\ell$ is Lipschitz smooth with respect to the model parameter $w$ with constant $L$, and have $\rho$-bounded gradients with respect to training/validation data. And the $\mathcal{A}(\theta)$ is differential with a $\delta$-bounded gradient and twice differential with its Hessian bounded by $\mathcal{B}$. Let the learning rate $\alpha_t = \mathcal{A}(\theta_t)$ predicted by MLR-SNet obey the exponential LR, i.e., $\alpha_t = \alpha_0 \beta^t, \beta = (\Gamma/T)^{1/T}, \Gamma \geq 1$. Let $\eta_t = \eta$ for all $t \in [T]$. If we use Adam algorithm to update MLR-SNet, we choose $\eta$ satisfied $\eta \leq \frac{\epsilon}{2L}$ and $1 - \beta_2 \leq \frac{\epsilon^2}{16\rho^2}$, where $\beta_2, \epsilon$ are the hyperparameter of the Adam algorithm. Then for $\theta_t$ generated using Adam, we have the following bound:*

$$\min_{0 \leq t \leq T} \mathbb{E}[\|\nabla \mathcal{L}_{Val}(\hat{\mathbf{w}}_t(\theta_t))\|_2^2] \leq \mathcal{O}(\frac{C \ln(T)}{T} + \sigma^2), \tag{19}$$

*where $C$ is some constant independent of the convergence process, $\sigma$ is the variance of drawing uniformly mini-batch sample at random.*

**Proof** *Suppose we have a small validation set with $M$ samples $\{x_1, x_2, \cdots, x_M\}$, each associating with a validation loss function $\ell_i(w(\theta))$, where $w$ is the parameter of the model, and $\theta$ is the parameter of the MLR-SNet. The overall validation loss would be,*

$$\mathcal{L}_{Val}(w) = \frac{1}{M} \sum_{i=1}^{M} \ell_i(w(\theta)). \tag{20}$$

*According to the updating algorithm 1, we have:*

$$\mathcal{L}_{Val}(\hat{w}_{t+1}(\theta_{t+1})) - \mathcal{L}_{Val}(\hat{w}_t(\theta_t))$$
$$= \underbrace{\{\mathcal{L}_{Val}(\hat{w}_{t+1}(\theta_{t+1})) - \mathcal{L}_{Val}(\hat{w}_t(\theta_{t+1}))\}}_{(a)} + \underbrace{\{\mathcal{L}_{Val}(\hat{w}_t(\theta_{t+1})) - \mathcal{L}_{Val}(\hat{w}_t(\theta_t))\}}_{(b)} \tag{21}$$

*For term (a),*

$$\mathcal{L}_{Val}(\hat{w}_{t+1}(\theta_{t+1})) - \mathcal{L}_{Val}(\hat{w}_t(\theta_{t+1}))$$
$$\leq \langle \nabla_w \mathcal{L}_{Val}(\hat{w}_{t+1}(\theta_{t+1})), \hat{w}_{t+1}(\theta_{t+1}) - \hat{w}_t(\theta_{t+1}) \rangle + \frac{L}{2} \|\hat{w}_{t+1}(\theta_{t+1}) - \hat{w}_t(\theta_{t+1})\|_2^2 \tag{22}$$

*According to Eq (6), we have*

$$\hat{w}_{t+1}(\theta_{t+1}) - \hat{w}_t(\theta_{t+1}) = -\alpha_t \nabla_w \mathcal{L}_{Tr}^B(\hat{w}_t(\theta_{t+1})) \tag{23}$$

*where $\alpha_t = \mathcal{A}(\mathcal{L}_{Tr}^B(\hat{w}_t(\theta_{t+1}); \theta_t), \mathcal{L}_{Tr}^B(w_t) = \frac{1}{|B_t|} \sum_{i \in B_t} \nabla \mathcal{L}_i^{Tr}(w_t)$. This can be written as*

$$\hat{w}_{t+1}(\theta_{t+1}) - \hat{w}_t(\theta_{t+1}) = -\alpha_t \left[ \nabla_w \mathcal{L}_{Tr}(\hat{w}_t(\theta_{t+1})) + \xi^{(t)} \right], \tag{24}$$

*where $\xi^{(t)} = \nabla_w \mathcal{L}_{Tr}^B(\hat{w}_t(\theta_{t+1})) - \nabla_w \mathcal{L}_{Tr}(\hat{w}_t(\theta_{t+1}))$. Since $B_t$ is the mini-batch samples drawn uniformly from the entire data set, we have $\mathbb{E}[\xi^{(t)}] = 0$. Furthermore, $\xi^{(t)}$ are i.i.d random variable with finite variance, since $B_t$ are drawn i.i.d with a finite number of samples. Then Eq (22) can be written as*

$$a \leq \left\langle \nabla_w \mathcal{L}_{Val}(\hat{w}_t(\theta_{t+1})), -\alpha_t \left[ \nabla_w \mathcal{L}_{Tr}(\hat{w}_t(\theta_{t+1})) + \xi^{(t)} \right] \right\rangle + \frac{L}{2} \left\| -\alpha_t \left[ \nabla_w \mathcal{L}_{Tr}(\hat{w}_t(\theta_{t+1})) + \xi^{(t)} \right] \right\|_2^2$$
$$= \left\langle \nabla_w \mathcal{L}_{Val}(\hat{w}_t(\theta_{t+1})), -\alpha_t \left[ \nabla_w \mathcal{L}_{Tr}(\hat{w}_t(\theta_{t+1})) + \xi^{(t)} \right] \right\rangle$$
$$+ \frac{L\alpha_t^2}{2} \left[ \|\nabla_w \mathcal{L}_{Tr}(\hat{w}_t(\theta_{t+1}))\|^2 + \|\xi^{(t)}\|_2^2 - \langle \nabla_w \mathcal{L}_{Tr}(\hat{w}_t(\theta_{t+1})), \xi^{(t)} \rangle \right]$$
$$\leq \left\langle \nabla_w \mathcal{L}_{Val}(\hat{w}_t(\theta_{t+1})), -\alpha_t \left[ \nabla_w \mathcal{L}_{Tr}(\hat{w}_t(\theta_{t+1})) + \xi^{(t)} \right] \right\rangle + \frac{L}{2}\alpha_t^2 \left[ \rho^2 + \|\xi^{(t)}\|_2^2 - \langle \nabla_w \mathcal{L}_{Tr}(\hat{w}_t(\theta_{t+1})), \xi^{(t)} \rangle \right].$$

*For term (b), according to Lemma 1, i.e., the validation loss is Lipschitz smooth with respect to the MLR-SNet parameter $\theta$, for briefly denote $L$.*

$$\mathcal{L}_{Val}(\hat{w}_t(\theta_{t+1})) - \mathcal{L}_{Val}(\hat{w}_t(\theta_t))$$
$$\leq \langle \nabla_\theta \mathcal{L}_{Val}(\hat{w}_t(\theta_t)), \theta_{t+1} - \theta_t \rangle + \frac{L}{2} \|\theta_{t+1} - \theta_t\|_2^2 \tag{25}$$

*If we adopt Adam to update the parameter of MLR-SNet, $\theta_{t+1} - \theta_t$ in Eq.(25) is updated by*

$$\theta_{t+1} = \theta_t - \eta_t \frac{g_{t,i}}{\sqrt{v_{t,i}} + \epsilon}, \tag{26}$$

*where $g_{t,i} = \nabla_\theta \mathcal{L}_{Val}^i(\hat{w}_t(\theta_t))$. Now, we have*

$$\mathcal{L}_{Val}(\hat{w}_t(\theta_{t+1})) - \mathcal{L}_{Val}(\hat{w}_t(\theta_t))$$
$$\leq -\eta_t \sum_{i=1}^d \left\langle \nabla_\theta \mathcal{L}_{Val}^i(\hat{w}_t(\theta_t)), \frac{g_{t,i}}{\sqrt{v_{t,i}} + \epsilon} \right\rangle + \frac{L\eta_t^2}{2} \sum_{i=1}^d \frac{g_{t,i}^2}{(\sqrt{v_{t,i}} + \epsilon)^2} \tag{27}$$

*Based on the proof process in (Zaheer et al., 2018) (Eq 4 in p. 13),*

$$\mathcal{L}_{Val}(\hat{w}_t(\theta_{t+1})) - \mathcal{L}_{Val}(\hat{w}_t(\theta_t))$$

$$\leq -\frac{\eta_t}{2(\sqrt{\beta_2}\rho + \epsilon)}\|\nabla_\theta \mathcal{L}_{Val}(\hat{w}_t(\theta_t))\|_2^2 + \left(\frac{\eta_t \rho\sqrt{1-\beta_2}}{\epsilon^2} + \frac{L\eta^2}{2\epsilon^2}\right)\frac{\sigma^2}{M}, \qquad (28)$$

*Now Eq.(21) has become the following:*

$$\mathcal{L}_{Val}(\hat{w}_{t+1}(\theta_{t+1})) - \mathcal{L}_{Val}(\hat{w}_t(\theta_t)) \leq \left\langle \nabla_w \mathcal{L}_{Val}(\hat{w}_t(\theta_{t+1})), -\alpha_t \left[\nabla_w \mathcal{L}_{Tr}(\hat{w}_t(\theta_{t+1})) + \xi^{(t)}\right]\right\rangle$$

$$+\frac{L}{2}\alpha_t^2 \left[\rho^2 + \|\xi^{(t)}\|_2^2 - \langle\nabla_w \mathcal{L}_{Tr}(\hat{w}_t(\theta_{t+1})), \xi^{(t)}\rangle\right] - \frac{\eta_t}{2(\sqrt{\beta_2}\rho + \epsilon)}\|\nabla_\theta \mathcal{L}_{Val}(\hat{w}_t(\theta_t))\|_2^2 + \left(\frac{\eta_t \rho\sqrt{1-\beta_2}}{\epsilon^2} + \frac{L\eta^2}{2\epsilon^2}\right)\frac{\sigma^2}{M},$$

$$(29)$$

*Taking expectations with respect to $\xi$ on both side of Eq.(29) and rearranging the inequality, we can obtain:*

$$\mathbb{E}_\xi \left[\frac{\eta_t}{2(\sqrt{\beta_2}\rho + \epsilon)}\|\nabla_\theta \mathcal{L}_{Val}(\hat{w}_t(\theta_t))\|_2^2\right]$$

$$\leq \alpha_t \rho^2 + \frac{L}{2}\alpha_t^2(\rho^2 + \sigma^2) - \mathcal{L}_{Val}(\hat{w}_{t+1}(\theta_{t+1})) + \mathcal{L}_{Val}(\hat{w}_t(\theta_t)) + \left(\frac{\eta_t \rho\sqrt{1-\beta_2}}{\epsilon^2} + \frac{L\eta^2}{2\epsilon^2}\right)\frac{\sigma^2}{M}$$

*Using telscoping sum, we obtain*

$$\sum_{t=1}^T \frac{\eta_t}{2(\sqrt{\beta_2}\rho + \epsilon)}\mathbb{E}\|\nabla_\theta \mathcal{L}_{Val}(\hat{w}_t(\theta_t))\|_2^2$$

$$\leq \mathcal{L}_{Val}(\hat{w}_t(\theta_1)) - \mathcal{L}_{Val}(\hat{w}_t(\theta_{T+1})) + \rho^2 \sum_{t=1}^T \alpha_t + \frac{L}{2}(\rho^2 + \sigma^2)\sum_{t=1}^T \alpha_t^2 + \left(\frac{\eta_t \rho\sqrt{1-\beta_2}}{\epsilon^2} + \frac{L\eta^2}{2\epsilon^2}\right)\frac{\sigma^2 T}{M}$$

$$\leq \mathcal{L}_{Val}(\hat{w}_t(\theta_1)) + \rho^2 \sum_{t=1}^T \alpha_t + \frac{L}{2}(\rho^2 + \sigma^2)\sum_{t=1}^T \alpha_t^2 + \left(\frac{\eta_t \rho\sqrt{1-\beta_2}}{\epsilon^2} + \frac{L\eta^2}{2\epsilon^2}\right)\frac{\sigma^2 T}{M}$$

$$(30)$$

*Therefore,*

$$\min_t \mathbb{E}_\xi \left[\|\nabla_\theta \mathcal{L}_{Val}(\hat{w}_t(\theta_t))\|_2^2\right] \leq \frac{\sum_{t=1}^T \frac{\eta_t}{2(\sqrt{\beta_2}\rho + \epsilon)}\mathbb{E}_\xi \left\|\nabla_\theta \mathcal{L}_{Val}(\hat{w}_t(\theta^{(t)}))\right\|_2^2}{\sum_{t=1}^T \frac{\eta_t}{2(\sqrt{\beta_2}\rho + \epsilon)}}$$

$$\leq \frac{\mathcal{L}_{Val}(\hat{w}_t(\theta_1)) + \rho^2 \sum_{t=1}^T \alpha_t + \frac{L}{2}(\rho^2 + \sigma^2)\sum_{t=1}^T \alpha_t^2 + \left(\frac{\eta_t \rho\sqrt{1-\beta_2}}{\epsilon^2} + \frac{L\eta^2}{2\epsilon^2}\right)\frac{\sigma^2 T}{M}}{\sum_{t=1}^T \eta_t} 2(\sqrt{\beta_2}\rho + \epsilon)$$

$$\leq \frac{2(\sqrt{\beta_2}\rho + \epsilon)}{T\eta}\left\{\mathcal{L}_{Val}(\hat{w}_t(\theta_1)) + \rho^2 \sum_{t=1}^T \alpha_t + \frac{L}{2}(\rho^2 + \sigma^2)\sum_{t=1}^T \alpha_t^2 + \left(\frac{\eta_t \rho\sqrt{1-\beta_2}}{\epsilon^2} + \frac{L\eta^2}{2\epsilon^2}\right)\frac{\sigma^2 T}{M}\right\}$$

$$\leq \mathcal{O}(\frac{\ln(T)}{T} + \sigma^2).$$

```
class LSTMCell(nn.Module):
def __init__(self, num_inputs, hidden_size):
super(LSTMCell, self).__init__()
self.hidden_size = hidden_size
self.fc_i2h = nn.Sequential(
nn.Linear(num_inputs, hidden_size),
nn.ReLU(),
nn.Linear(hidden_size, 4 * hidden_size)
)
self.fc_h2h = nn.Sequential(
nn.Linear(hidden_size, hidden_size),
nn.ReLU(),
nn.Linear(hidden_size, 4 * hidden_size)
)

def forward(self, inputs, state):
hx, cx = state
i2h = self.fc_i2h(inputs)
h2h = self.fc_h2h(hx)
x = i2h + h2h
gates = x.split(self.hidden_size, 1)
in_gate = torch.sigmoid(gates[0])
forget_gate = torch.sigmoid(gates[1])
out_gate = torch.sigmoid(gates[2])
in_transform = torch.tanh(gates[3])
cx = forget_gate * cx + in_gate * in_transform
hx = out_gate * torch.tanh(cx)
return hx, cx

class MLRNet(nn.Module):
def __init__(self, num_layers, hidden_size):
super(MLRNet, self).__init__()
self.hidden_size = hidden_size
self.layer1 = LSTMCell(1, hidden_size)
self.layer2 = nn.Linear(hidden_size, 1)

def forward(self, x, gamma):
self.hx, self.cx = self.layer1(x, (self.hx, self.cx))
x = self.hx
x = self.layer2(x)
out = torch.sigmoid(x)
return gamma * out
```

# B  EXPERIMENTAL DETAILS AND ADDITIONAL RESULTS IN SECTION 4.1

In this section, we attempt to evaluate the capability of MLR-SNet to learn LR schedules compared with baseline methods. Here, we provide implementation details of all experiments.

**Datasets.** We choose two datasets in image classification (CIFAR-10 and CIFAR-100), and one dataset in text classification (Penn Treebank) to present the efficiency of our method. CIFAR-10 and CIFAR-100 Krizhevsky (2009), consisting of $32 \times 32$ color images arranged in 10 and 100 classes, respectively. Both datasets contain 50,000 training and 10,000 test images. Penn Treebank Marcus & Marcinkiewicz is composed of 929k training words, 73k validation words, and 82k test words, with a 10k vocabulary in total. Our algorithm and RTHO Franceschi et al. (2017) randomly select 1,000 clean images in the training set of CIFAR-10/100 as validation data, and directly use the validation set in Penn Treebank as validation data.

**CIFAR-10 & CIFAR-100.** We employ ResNet-18 on CIFAR-10 and WideResNet-28-10 Zagoruyko & Komodakis (2016) on CIFAR-100. All compared methods and MLR-SNet are trained for 200 epochs with batch size 128. For baselines involving SGD as base optimizer, we set the initial LR to 0.1, weight decay parameter to $5e^{-4}$ and momentum to 0.9 if used. While for **Adam**, we just follow

Table 2: Test accuracy (%) of CIFAR dataset with SGD baselines.

| Optimizer | CIFAR-10 with ResNet18 | CIFAR-100 with WRN-28-10 |
|---|---|---|
| SGD+Fixed | $92.26 \pm 0.12$ | $70.67 \pm 0.34$ |
| SGD+MultiStep | $93.82 \pm 0.09$ | $77.04 \pm 0.17$ |
| SGD+Exponential | $90.93 \pm 0.11$ | $72.52 \pm 0.34$ |
| SGD+SGDR | $93.92 \pm 0.11$ | $72.52 \pm 0.34$ |
| Adam | $90.86 \pm 0.15$ | $68.94 \pm 0.24$ |
| SGD+L4 | $89.15 \pm 0.14$ | $63.61 \pm 0.65$ |
| SGD+HD | $92.34 \pm 0.09$ | $72.22 \pm 0.30$ |
| SGD+RTHO | $92.60 \pm 0.18$ | $72.32 \pm 0.47$ |
| MLR-SNet (Meta-train) | $94.70 \pm 0.16$ | $79.41 \pm 0.14$ |

Table 3: Test accuracy (%) of CIFAR dataset with SGDM baselines.

| Optimizer | CIFAR-10 with ResNet18 | CIFAR-100 with WRN-28-10 |
|---|---|---|
| SGDM+Fixed | $87.69 \pm 0.14$ | $70.88 \pm 0.12$ |
| SGDM+MultiStep | $95.08 \pm 0.13$ | $80.74 \pm 0.19$ |
| SGDM+Exponential | $94.64 \pm 0.05$ | $78.87 \pm 0.04$ |
| SGDM+SGDR | $95.06 \pm 0.17$ | $80.93 \pm 0.05$ |
| Adam | $90.86 \pm 0.15$ | $68.94 \pm 0.24$ |
| SGDM+L4 | $91.03 \pm 0.14$ | $66.51 \pm 2.83$ |
| SGDM+HD | $93.99 \pm 0.12$ | $76.80 \pm 0.19$ |
| SGDM+RTHO | $93.17 \pm 0.49$ | $76.14 \pm 0.29$ |
| MLR-SNet (Meta-train) | $94.70 \pm 0.16$ | $79.41 \pm 0.14$ |

the default parameter setting. The hyper-parameters of hand-designed LR schedules are listed below: **Exponential** decay, multiplying LR with $0.95$ every epoch; **MultiStep** decay, decaying LR by 10 every 60 epochs; **SGDR**, setting T_0 to 10, T_Mult to 2 and minimum LR to $1e^{-5}$. **L4**, **HD** and **RTHO** update LR every data batch, and we use the recommended setting in the original paper of **L4** ($\alpha = 0.15$) and search different hyper-lrs from $\{1e^{-3}, 1e^{-4}, 1e^{-5}, 1e^{-6}, 1e^{-7}\}$ for **HD** and **RTHO**, reporting the best performing hyper-lr.

**Penn Treebank.** We use a 2-layer and 3-layer LSTM network which follows a word-embedding layer and the output is fed into a linear layer to compute the probability of each word in the vocabulary. Hidden size of LSTM cell is set to $512$ and so is the word-embedding size. We tie weights of the word-embedding layer and the final linear layer. Dropout is applied to the output of word-embedding layer together with both the first and second LSTM layers with a rate of $0.5$. As for training, the LSTM net is trained for 150 epochs with a batch size of 32 and a sequence length of 35. We set the base optimizer SGD to have an initial LR of 20 without momentum, for Adam, the initial LR is set to $0.01$ and weight for moving average of gradient is set to 0. We apply a weight decay of $5e^{-6}$ to both base optimizers. All experiments involve a $0.25$ clipping to the network gradient norm. For both SGD and Adam, we decrease LR by a factor of 4 when performance on validation set shows no progress. For L4, we try different $\alpha$ in $\{0.1, 0.05, 0.01, 0.005\}$ and reporting the best test perplexity among them. For both **HD** and **RTHO**, we search the hyper-lr lying in $\{1, 0.5, 0.1, 0.05\}$, and report the best results.

Table 4: Test perplexity on the Penn Treebank dataset.

| Optimizer | 2-layer LSTM | 3-layer LSTM |
|---|---|---|
| SGD+Val Strategy | $74.33 \pm 0.23$ | $76.05 \pm 0.39$ |
| Adam+Val Strategy | $71.17 \pm 0.23$ | $74.80 \pm 0.73$ |
| SGD+L4 | $82.58 \pm 1.32$ | $92.27 \pm 0.92$ |
| SGD+HD | $76.90 \pm 0.33$ | $78.63 \pm 0.08$ |
| SGD+RTHO | $76.69 \pm 0.11$ | $78.52 \pm 0.16$ |
| MLR-SNet (Meta-train) | $72.09 \pm 0.72$ | $72.71 \pm 0.17$ |

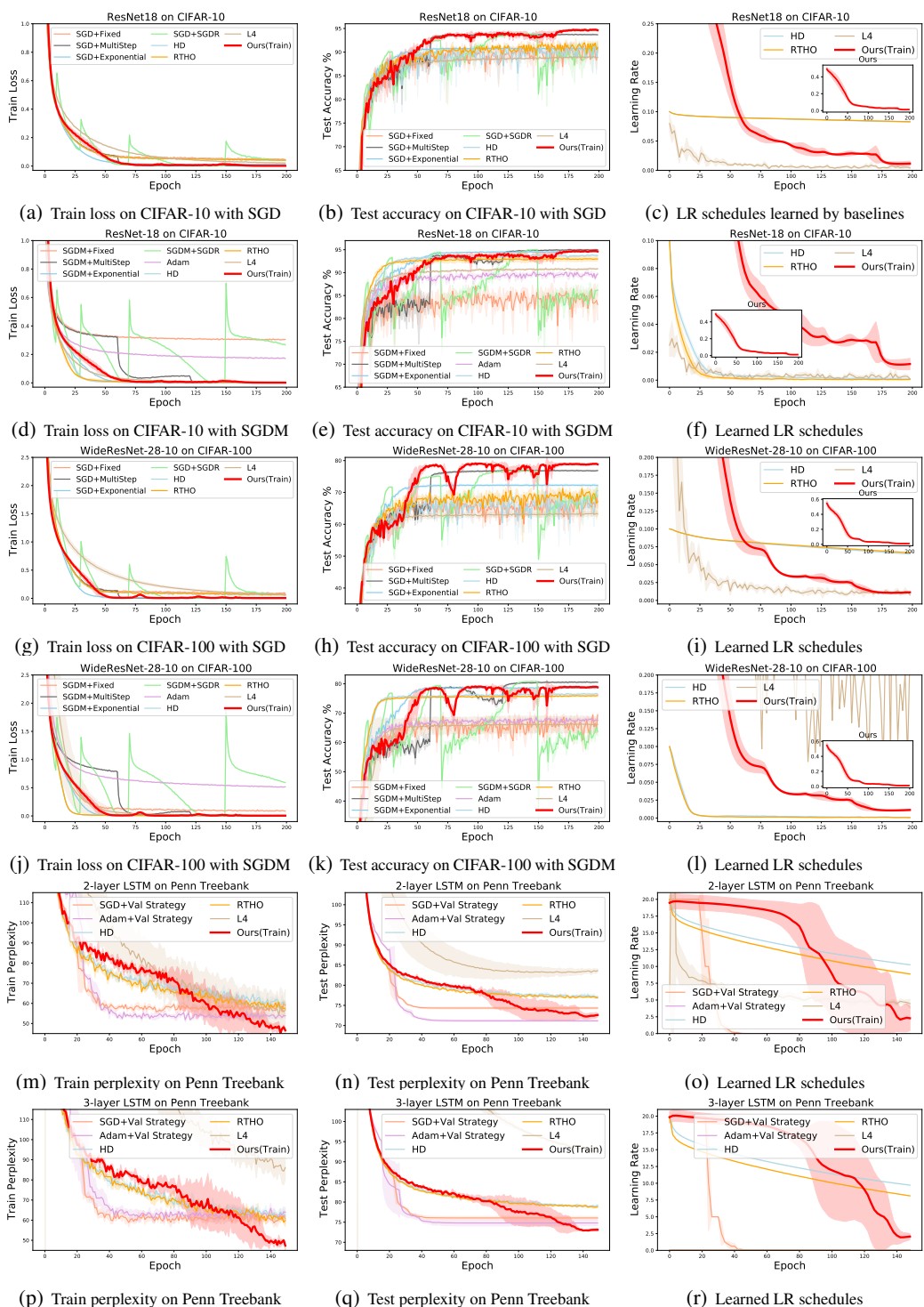

Figure 11: Train loss (perplexity), test accuracy (perplexity) and learned LR schedules of our methods (train) and compared baselines on different tasks.

**MLR-SNet architecture and parameter setting.** The architecture of MLR-SNet is illustrated in Section 3.2. In our experiment, the size of hidden nodes is set as 40. The initialization of MLR-SNet follows the default setting in Pytorch. The Pytorch implementation of MLR-SNet is listed above.

We employ Adam optimizer to train MLR-SNet, and just set the parameters as originally recommended with a LR of $1e^{-3}$, and a weight decay of $1e^{-4}$, which avoids extra hyper-parameter tuning. For image classification tasks, the input of MLR-SNet is the training loss of a mini batch samples. Every data batch's LR is predicted by MLR-SNet and we update it twice per epoch according to the loss of the validation data. While for text classification tasks, we take $\frac{\mathcal{L}_{Tr}}{\log(vocabulary\ size)}$ as input of MLR-SNet to deal with the influence of large scale classes of text. MLR-SNet is updated every 100 batches due to the large number of batches per epoch compared to that in image datasets.

**Results.** Due to the space limitation, we only present the test accuracy in the main paper. Here, we present the training loss and test accuracy of our method and all compared methods on image and text tasks, as shown in Fig.11. For image tasks, except for Adam and SGD with fixed LR, other methods can decrease the loss to 0 almostly. Though local minima can be reached by these methods, the generalization ability of the these mimimas has a huge difference, which can be summarized from test accuracy curves. As shown in Fig. 11(a),11(b),11(g),11(h), when using SGD to train DNNs, the compared methods SGD with Exponential LR, L4, HD, RTHO fail to find such good solutions to generalize well. Especially, L4 greedily searches LR to decrease loss to 0, making it fairly hard to adapt the complex DNNs training dynamic and obtain a good mimima, while our method can adjust LR to comply with the significant variations of training dynamic, leading to a better generalization solution. As shown in Fig. 11(d),11(e),11(j),11(k), when baseline methods are trained with SGDM, these methods make a great progress in escaping from the bad minimas. In spite of this, our method still shows superiority in finding a solution with better generalization compared with these competitive training strategies.

In the third column in Fig. 11, we plot learned LR schedules of compared methods and our method. As can be seen, our method can learn LR schedules approximating the hand-designed LR schedules while with more locally varying. HD and RTHO often have the same trajectory while producing lower or faster downward trend than ours. This tends to explain our final performances on test set is better than HD and RTHO, since our method can adaptively adjust LR utilizing the past training histories explicitly. L4 greedily searches a LR to decrease the loss. This often leads to a large value causing fluctuations or even divergence (Fig. 11(l)), or a small value causing slow progress (Fig. 11(r)), or both of them (Fig. 11(c) 11(f) 11(i) 11(o)). Such LR schedules often result in bad mimimas. Moreover, all compared methods regard LR as hyper-parameter to learn without a transferable formulation, and the learned LR schedules can not generalize to other learning tasks directly. Generally, they just try to find a proper LR schedule from scratch for new tasks. However, our meta-learned MLR-SNet is plug-and-play and transferrable, which can directly transfer how to schedule LR for SGD to heterogeneous tasks without additional learning.

**Ablation study.**

(1) **The architecture of MLR-SNet.** Fig.12(a) shows the test accuracy on CIFAR-10 with ResNet-18 of different architectures of MLR-SNet. As can be seen, our algorithm is not sensitive to the choose of the MLR-SNet's architectures. This implies that our algorithm is robust and stable for helping improve DNN training.

(2) **The gobal LR of the meta optimizer**. To further validate that whether our MLR-SNet behaves robust to the meta optimizer. We adapot Adam optimizer to search the proper LR schedules. Fig. 12(b) shows that our MLR-SNet achieves the similar performance even for different global LRs. This implies our MLR-SNet needs not carefully tune the LR of the meta optimizer, which makes it easy to reproduce and apply to various problems.

(3) **The different $\gamma$ values of the MLR-SNet**. One important hyperparameter of the MLR-SNet is $\gamma$, here we verify our method is not sensitive to the choose of $\gamma$ value. We test $\gamma$ values from $0.1$ to $10$ for the DNNs training. As shown in Fig.12(c), even with different learning scales, our method can still help DNNs achieve almost similar performance. This implies the MLR-SNet is robust to the choose of the $\gamma$, which makes it easy to be applied into parctice.

## C   EXPERIMENTAL DETAILS AND ADDITIONAL RESULTS IN SECTION 4.2

We investigate the transferability of the learned LR schedule when applied to various tasks in Section 4.2 of the main paper. We use the MLR-SNet meta-learned on CIFAR-10 with ResNet-18 in Section 4.1 to directly predict the LR for SGD algorithm to new heterogeneous tasks. We save the learned MLR-SNet at different epochs in the whole one meta-train run. As is shown in Fig.13(a), if we use

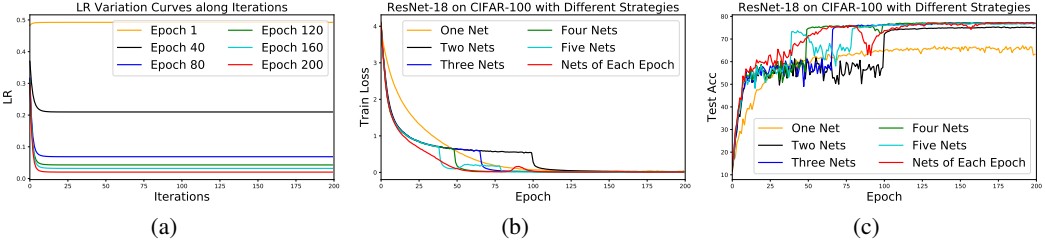

(a) Different Architectures of MLR-(b) Different LRs of Meta Optimizer (c) Different $\gamma$ value of MLR-SNet
SNet

Figure 12: Ablation study. (a) Test accuracy on CIFAR-10 with ResNet-18 of different architectures of MLR-SNet. 'a-b' denotes the configurations of MLR-SNet, where 'a' represents the number of layers, and 'b' represents the number of hidden nodes. (b)Test accuracy on CIFAR-10 with ResNet-18 of different LRs of meta optimizer 'Adam'. (c) Test accuracy on CIFAR-10 with ResNet-18 of different $gamma$ values of MLR-SNet.

Figure 13: (a) We plot the LR variation curves along iterations with the same input for learned MLR-SNet at different epochs. As is shown, when iteration increases, the LR is almost constant. This means the learned MLR-SNet overfits the short trajectories, while fails for the long trajectories. (b),(c) show the recording train loss and test accuracy with ResNet-18 on CIFAR-100 of different test strategies.

the single learned MLR-SNet at certain epoch, it can be seen that the predicted LR by the learned LR schedules converges after several iterations. This is because that the training trajectories are long in our experiments, and the learned MLR-SNet can not memory all the information since we locally adjust our MLR-SNet according to the validation error. If we directly select one MLR-SNet learned at any epoch, that will raise overfitting issues as shown in Fig.13(a). Thus we should select more than two learned MLR-SNets for test. Here, we propose a heuristic strategy to select MLR-SNets for test. Generally, if we want to select $k$ nets for test, the MLR-SNet learned at $\left\lceil\frac{200*l}{k-1}\right\rceil$-th epoch $(l = 0, 1, 2, \cdots, k-1)$ should be chosen, where $\lceil\cdot\rceil$ denotes ceiling operator. Fig.13(b) and 13(c) show the train loss and test accuracy with ResNet-18 on CIFAR-100 of different test strategies, i.e., choosing different number of nets to transfer. It can be seen that almost choosing more than three nets have similar performance. Therefore, in the following experiments we choose three MLR-SNets to show the transferability.

**Transfer to different epochs.** We transfer the LR schedules meta-trained with epoch 200 to other different epochs, e.g., 100, 400,1200. All the methods are trained with ResNet-18 on CIFAR-100 with batch size 128 for different epochs. The hyper-parameter setting for compared hand-designed LR schedules is the same with Section 4.1 in the main paper as illustrated above, except for MultiStep LR. For epoch 100, MultiStep LR decays LR by 10 every 30 epochs; For epoch 400, MultiStep LR decays LR by 10 every 120 epochs; For epoch 1200, MultiStep LR decays LR by 10 every 360

Table 5: Test accuracy (%) of CIFAR-10 dataset with different networks.

| Optimizer | ShuffleNetV2 | MobileNetV2 | NASNet |
|---|---|---|---|
| SGD+Fixed | $87.06 \pm 0.33$ | $90.85 \pm 0.52$ | $89.14\pm 1.15$ |
| SGD+MultiStep | $88.99 \pm 0.11$ | $92.28 \pm 0.28$ | $94.97 \pm 0.10$ |
| SGD+Exponential | $89.45 \pm 0.14$ | $93.47 \pm 0.12$ | $94.59 \pm 0.14$ |
| SGD+SGDR | $89.30 \pm 0.83$ | $92.00 \pm 1.21$ | $94.75 \pm 0.82$ |
| Adam | $87.95 \pm 0.31$ | $90.64 \pm 0.54$ | $90.60 \pm 0.47$ |
| MLR-SNet (Meta-test) | $89.09 \pm 0.33$ | $93.11 \pm 0.10$ | $95.18\pm0.18$ |

Table 6: Validation accuracies on ImageNet dataset.

| Optimizer | Top-1 Accuracy | Top-5 Accuracy |
|---|---|---|
| SGDM+Fixed | 68.23 | 88.47 |
| SGDM+MultiStep | 75.90 | 92.90 |
| SGDM+Exponential | 75.68 | 92.69 |
| SGDM+SGDR | 75.82 | 92.87 |
| Adam | 63.62 | 85.43 |
| MLR-SNet (Meta-test) | 75.03 | 92.39 |

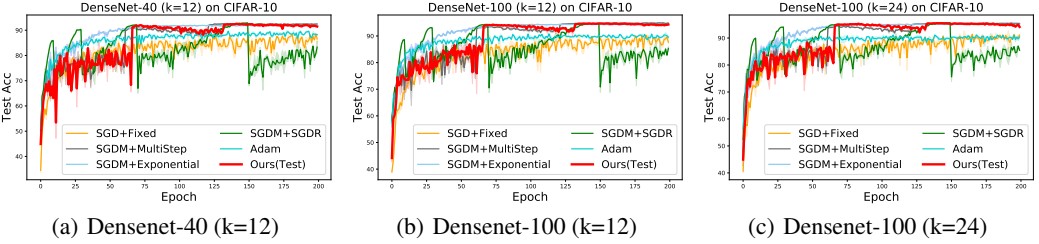

(a) Densenet-40 (k=12)      (b) Densenet-100 (k=12)      (c) Densenet-100 (k=24)

Figure 14: Test accuracy on CIFAR-100 of different DenseNet architectures.

epochs. Other hyper-parameters of MultiStep LR keep unchanged. For our method, we use the transferred strategy as below: 1) For epoch 100, we employ the 3 nets at 0-33, 33-67, 67-100 epoch, respectively; 2) For epoch 400, we employ the 3 nets at 0-133, 133-267, 267-400 epoch, respectively; 3) For epoch 1200, we employ the 3 nets at 0-400, 400-800, 800-1200 epoch, respectively.

**Transfer to different datasets.** We transfer the LR schedules meta-learned on CIFAR-10 to SVHN (Netzer et al., 2011), TinyImageNet [6], and Penn Treebank (Marcus & Marcinkiewicz). For image classification, we train a ResNet-18 on SVHN and TinyImageNet, respectively. The hyper-parameters of all compared methods are set the same as those of CIFAR-10. For text classification, we train a 3-layer LSTM on Penn Treebank. The hyper-parameters of all compared methods are with the same setting as introduced in Section 4.1.

**Transfer to different net architectures.** We transfer the learned LR schedules for different net architectures training. All the methods are trained on CIFAR-10 with different net architectures. The hyper-parameters of all methods are the same with the setting of CIFAR-10 with ResNet-18. We test the meta-learned LR schedule to different configurations of DenseNet Huang et al. (2017). As shown in Fig. 14, our method perform slightly stable than MultiStep strategy at about 75-125 epochs. This tends to show the superiority of adaptive LR to train the DenseNets. Also, we transfer the LR schedules to several novel networks, the results are presented in Fig.8 in the main paper.

**Transfer to large scale optimization.** We transfer the learned LR schedules for the training of the large scale optimization problems. The predicted LR by MLR-SNet will not substantially increase the complexity compared with hand-designed LR schedules for DNNs training. This makes it feasible and reliable to transfer our meta-learned LR schedules to such large scale optimization problems. We train a ResNet-50 on ImageNet with hand-designed LR schedules and our transferred LR schedules. The training code can be found on `https://github.com/pytorch/examples/tree/master/imagenet`, and the parameter setting keeps unchanged except the LR. All compared hand-designed LR schedules are trained by SGDM with a momentum 0.9, a weight decay $5e^{-4}$, an initial learning rate 0.1 for 90 epochs, and batch size 256. **Fixed** LR uses 0.1 LR during the whole training; **Exponential** LR multiplies LR with 0.95 every epoch; **MultiStep** LR decays LR by 10 every 30 epochs; **SGDR** sets T_0 to 10, T_Mult to 2 and minimum LR to $1e^{-5}$; **Adam** just uses the default parameter setting. The results are presented in Fig. 9 in the main paper.

---

[6]It can be downloaded at https://tiny-imagenet.herokuapp.com.

Table 7: Test accuracy (%) on CIFAR-10 and CIFAR-100 training set of different methods trained on CIFAR-10-C and CIFAR-100-C. Best and Last denote the results of the best and the last epoch. The **Bold** and **Underline Bold** denote the first and second best results, respectively.

| Datasets/Methods | | Fixed | MultiStep | Exponential | SGDR | Adam | Ours(Train) |
|---|---|---|---|---|---|---|---|
| CIFAR-10-C | Best | 79.96±4.09 | 85.64±1.71 | 83.63±1.38 | **86.10±1.44** | 81.57±1.39 | **85.73±1.71** |
| | Last | 77.89±4.05 | **85.48±1.71** | 83.47±1.37 | 78.46±1.92 | 80.39±1.65 | **85.62±1.76** |
| CIFAR-100-C | Best | 46.91±3.08 | 52.38±2.43 | 49.90±1.93 | **52.80±2.39** | 45.58±1.95 | **52.51±2.38** |
| | Last | 44.81±5.98 | **52.28±2.44** | 49.75±1.94 | 41.68±3.33 | 43.94±2.18 | **52.35±2.46** |

## D  EXPERIMENTAL DETAILS AND ADDITIONAL RESULTS IN SECTION 4.3

The datasets CIFAR-10-C and CIFAR-100-C Hendrycks & Dietterich (2019) can be downloaded at `https://zenodo.org/record/2535967#.Xt4mVigzZPY`, `https://zenodo.org/record/3555552#.Xt4mdSgzZPY`. Each dataset contains 15 types of algorithmically generated corruptions from noise, blur, weather, and digital categories. These corruptions contain Gaussian Noise, Shot Noise, Impulse Noise, Defocus Blur, Frosted Glass Blur, Motion Blur, Zoom Blur, Snow, Frost, Fog, Brightness, Contrast, Elastic, Pixelate and JPEG. All the corruptions are gererated on 10,000 test set images, and each corruption contains 50,000 images since each type of corruption has five levels of severity. We treat CIFAR-10-C or CIFAR-100-C dataset as training set, and train a model with ResNet-18 for each corruption dataset. Finally, we can obtain 15 models for CIFAR-10/100-C. Each corruption can be roughly regarded as a task, and the average accuracy of 15 models on test data [7] is used to evaluate the robust performance of different tasks for each LR schedules strategy.

For experimental setting in Section 4.3, all compared hand-designed LR schedules are trained with a ResNet-18 by SGDM with a momentum 0.9, a weight decay $5e^{-4}$, an initial learning rate 0.1 for 100 epochs, and batch size 128. **Fixed** LR uses 0.1 LR during the whole training; **Exponential** LR multiplies LR with 0.95 every epoch; **MultiStep** LR decays LR by 10 every 30 epochs; **SGDR** sets T_0 to 10, T_Mult to 2 and minimum LR to $1e^{-5}$; **Adam** just uses the default parameter setting. Our method trains the ResNet-18 by SGD with a weight decay $5e^{-4}$, and the MLR-SNet is learned under the guidance of a small set of validation set without corruptions. We randomly choose 10 clean images for each class as validation set. The experimental result is listed in Table 1 in the main paper.

**Additional robustness results of transferrable LR schedules on different data corruptions.** Furthermore, we want to explore the robust performance of different tasks for our transferrable LR schedules. Different from above experiments where all 15 models are trained under the guidance of a small set of validation set, we just train a ResNet-18 on Gaussian Noise corruption to meta-learn the MLR-SNet, and then transfer the meta-learned LR schedules to other 14 corruptions. We report the average accuracy of 14 models on test data to show the robust performance of our transferred LR schedules. All the methods are meta-tested with a ResNet-18 for 100 epochs with batch size 128. The hyper-parameter setting of hand-designed LR schedules keeps same with above. Table 7 shows the mean test accuracy of 14 models. As can be seen, our transferrable LR schedules obtain the final best performance compared with hand-designed LR schedules. This implies that our transferrable LR schedules can also perform robust and stable than the pre-set LR schedules when the learning tasks are changed. However, our transferrable LR schedules are plug-and-play, and have no additional hyper-parameters to tune when transferred to new heterogeneous tasks.

## E  THE PRELIMINARY EXPLORATION OF THE INFLUENCE ON META-TEST TASKS OF THE DIFFERENT META-TRAINING TASKS.

In this section, we study the performance influence on the target task of the different meta-training tasks. Here we fixed the target task as training ResNet-18 on TinyImageNet. We choose three different meta-training task: 1) training ResNet-18 on CIFAR-10; 2) training WideResNet-28-10 on CIFAR-100; 3) training 3-layer LSTM on Penn Treebank. Fig.15 shows the meta-test performance of three different meta-training tasks. It can be seen that the meta-training task more related to the target task would obtain better transferable performance on the target task.

---

[7]We use the original 50,000 train images of CIFAR-10/100 as test data.

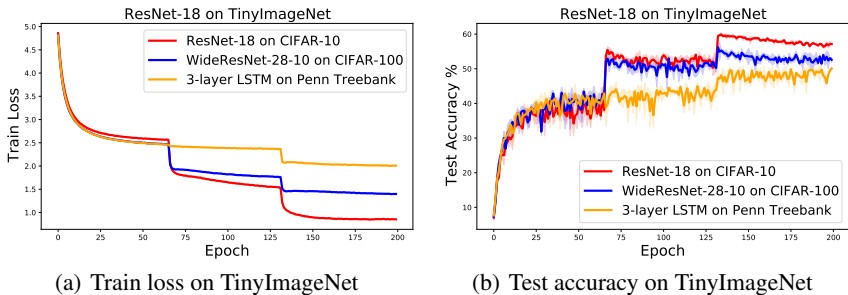

(a) Train loss on TinyImageNet       (b) Test accuracy on TinyImageNet

Figure 15: The meta-test performance on TinyImageNet (target task) of different meta-training tasks.

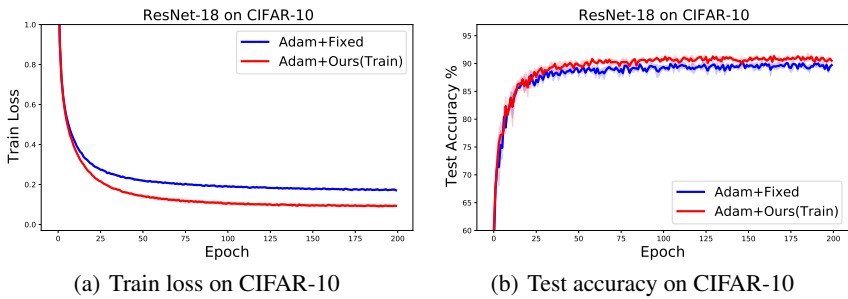

(a) Train loss on CIFAR-10       (b) Test accuracy on CIFAR-10

Figure 16: Performance comparion on CIFAR-10 of the MLR-SNet and baselines.

## F  APPLYING MLR-SNET ON TOP OF ADAM.

To further demostrate the versatility of our method, we apply the MLR-SNet on top of the Adam algorithm. Fig.16 shows that our methods can substantially improve the performance of the original Adam algorithm.

## G  EXPERIMENTAL RESULTS OF ADDITIONAL COMPARED METHOD LR CONTROLLER

In this section, we present the experimental results of LR Controller Xu et al. (2019), which is a related work of ours but under the reinforcement learning framework. Due to their learning algorithm is relatively computationally expensive and not very easy to optimize, we will show our method has a superiority in finding such a good LR schedule that scales and generalizes.

To start a fair comparison, we follow all the training settings and structure of LR Controller proposed in Xu et al. (2019) except that we modify the batch size to 128 and increase training steps to cover 200 epochs of data to match our setup in Section 4.1 [8]. Firstly, we train LR Controller on CIFAR-10 with ResNet-18 and CIFAR-100 with WideResNet-28-10 as we do in Section 4.1. As shown in Fig. 17, our method demonstrates evident superiority in finding a solution with better generalization compared with LR Controller strategies. LR Controller performs steadily in the early training phase, but soon fluctuates significantly and fails to progress. This tends to show that the LR Controller suffers from a severe stability issue when training step increases, especially being compared to our MLR-SNet.

Then we transfer the LR schedules learned on CIFAR-10 for our method and LR Controller to CIFAR-100 to verify their transferability. Test settings are the same with those related in Section 4.2. As shown in Fig. 18, LR Controller makes a comparatively slower progress in the whole training process. While our method achieves a competitive performance, which indicates the capability of transferring to other tasks for our method.

---

[8]Code for LR Controller can be found at https://github.com/nicklashansen/adaptive-learning-rate-schedule

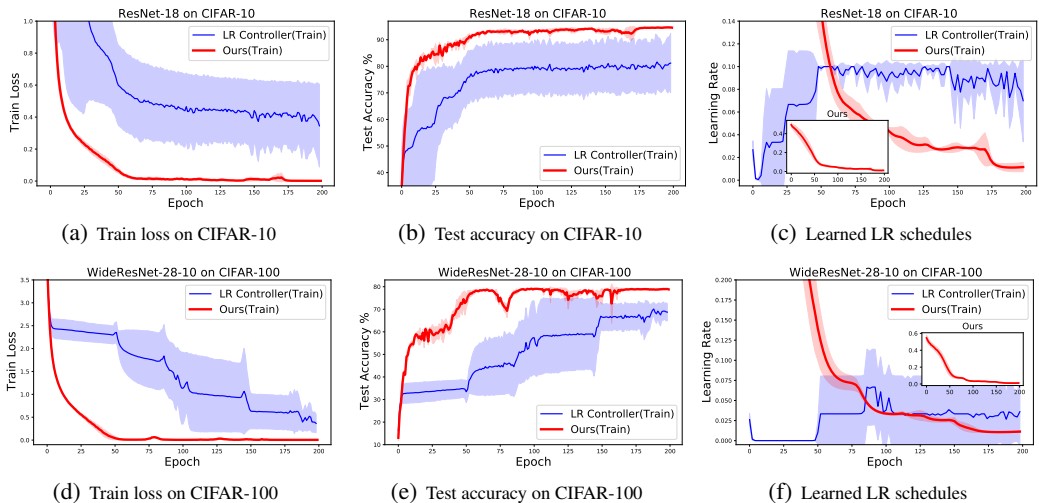

Figure 17: Train loss, test accuracy and learned LR schedules of our method(train) and LR Controller(train) on CIFAR-10 and CIFAR-100.

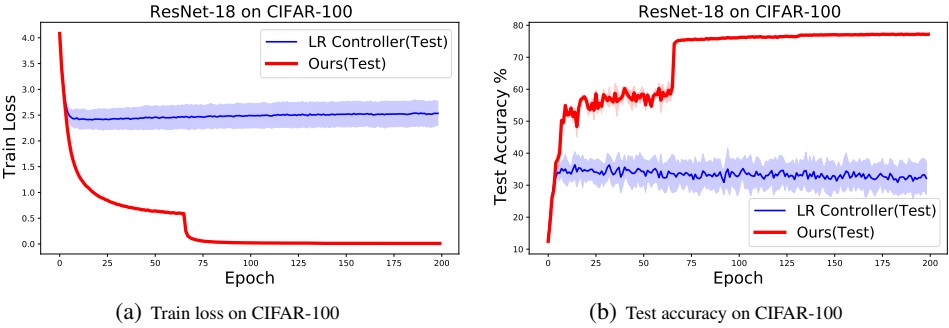

Figure 18: Train loss, test accuracy of our method(test) and LR Controller(test) on CIFAR-100.

