# OpenReview forum: "MLR-SNet: Transferable LR Schedules for Heterogeneous Tasks"
_ICLR.cc/2021/Conference — Reject_

### Official Review · AnonReviewer3 · 2020-10-27
**A practical and potentially high impact work for SGN based DNN training.**

**Rating:** 6
**Confidence:** 4

**Review:**

Summary: The paper proposes to parameterize learning rate (LR) schedule with an explicit mapping formulation. This learnable structure allowed the proposed meta-trained MLR-SNet to achieve good LR schedules. For validation, the proposed method is evaluated on both image and text classification benchmark with various network architectures and datasets, as well as transfer the learned network for new task or architectures.

Justification of rating: The paper solve a practical problem that is not handled in the existing literature. Despite the straightforwardness of the proposed approach and the methodology, this work has the potential to bring high impact to the research community.

Strengths:
+ This work proposed to parameterize the LR schedule with a MLR-SNet.  The results shows it is more flexible and general than the hand-picked LR schedule.
+ The meta-learned approach allow the learned model to be applied to unseen data.
+ The paper provide comprehensive experiment to validate the efficacy of the proposed model.

Comments:
- Experiment on Penn Treebank shows the convergence of the proposed MLR-SNet is slower than SGD and Adam. The paper argue that it predicts LR according to training dynamics by minimizing the validation loss. Please provide more details why is this a more intelligent way to employ validation set. Is this also observed in any other dataset?
- This work transfer the learned LR schedules on CIFAR-10 with ResNet-18 to several other datasets. Has the author  try to transfer MLR-SNet learned with other source dataset? How will the model trained with different model/dataset behave when transferred to new datasets or networks. It might be interesting to explore if the certain type of network architecture (more complex or simple) would learned a more generic model.
- In the "Formulation of MLR-SNet", it states that the input $h_{t-1}$ and the training loss are preprocessed by a fully-connected layer $W_1$ with ReLu activation function. Please describe the purpose of this layer.

---

> ### Author Response · Authors · 2020-11-18
> **Response to Reviewer #3**
>
> Thank you for the constructive comments! We really appreciate the comments for improving the clarity of statements and experimental verifications.
>
> Q3.1 More intelligent way to employ validation set.
>
> A.	Thanks for your question. The hand-designed LR schedules drop the LR by a fixed factor when the validation loss stops decreasing. This strategy only uses the loss information on validation set, and searches LR in a limited range. Our method optimizes the parameters of MLR-SNet by minimizing the validation loss, which uses the both loss and gradient information, and searches LR in a continuous real space (more freedom and flexibility). This tends to be more accurate and efficient. Meanwhile, the validation (meta) loss pushes the MLR-SNet towards the better generalization solutions involving in the optimization process.
>
>
> Q3.2 Ablation study on the influence of the meta-training tasks.
>
> A.	Thanks for your suggestion. We conduct additional experiments in the updated version. The preliminary experimental results [Fig.15 in Appendix E] shows the task relatedness have a slight impact on the transferable performance.
>
>
>
> Q3.3 The purpose of the fully-connected layer.
>
> A.	The additional fully-connected layer and ReLU activation function try to enhance the nonlinearity of the MLR-SNet to guarantee that it can fit more complex training dynamics.

---

### Official Review · AnonReviewer4 · 2020-10-28
**Interesting paper proposing black box generated learning rate schedules.**

**Rating:** 6
**Confidence:** 4

**Review:**

In this work, the authors use an LSTM to meta-learn learning rate schedules. This LSTM depends only on the validation loss at time t. They train this LSTM for some tasks and they show that it gives good performance when compared with baselines. Then they transfer one of these trained LSTMs to different tasks and gives good results.

Understanding learning rate schedules is an interesting problem in machine learning and the authors seem to provide a useful blackbox learning rate scheduler.

-In the SM, they discuss that they have to choose 3 MLR-SNET for transferability. This part is very confusing and should be written more clearly. This should also be mentioned in the main text because the impression there is that one just transfered the MLR-SNET as is.
-How many runs are needed to meta train the LSTM? Is the transferable LSTM coming from only one run?

-How robust is the performance of MLR-SNET with respect to different initial learning rates?

-It would be nice if the authors could add tables reporting the accuracies, it is not clear from the plots how good/bad the performance of MLR-SNET is. The Adam baseline is pretty weak, it should be supplemented with learning rate schedules, one should be able to get acc > 73% with Adam+schedules on imagenet.

---

> ### Author Response · Authors · 2020-11-18
> **Response to Reviewer #4**
>
> Thank you for the constructive comments! We really appreciate the comments for improving the clarity of statements and experimental verifications.
>
> Q4.1 About the description of the transferability experiment?
>
> A.	Thanks for your suggestion. We will write more clearly in revision. In the meta-train stage (Algorithm1), the train runs is same with the baselines methods, and we save the learned MLR-SNet at different epochs in the whole one meta-train run.
>
> Q4.2 How robust is the performance of MLR-SNet with respect to different initial learning rates?
>
> A.	Thanks for your question. In our understanding, the initial LR may contain two aspects. One is the LR of the meta-optimizer, i.e., the global LR of the Adam optimizer. We provide an ablation study of this point with varying LR from 5e-4 to 1e-3. The experimental results [Fig12(b) in Appendix B] shows our method is robust to such global LR. The other is the initial LR predicted by MLR-SNet for DNNs training. This LR is influenced by the $\gamma$. We conduct additional ablation experiments in the updated version. The experimental results [Fig12(c) in Appendix B] shows our method is also robust to varying $\gamma$ value.
>
>
>
> Q4.3 Reporting the accuracies and the Adam results on ImageNet.
>
> A.	Thanks for your suggestion. We will add tables reporting the accuracies in revision. The Adam baseline is used with fixed global LR 1e-3. We will attempt to add other LR schedules to get acc > 73% on ImageNet in revision.

---

> > ### Comment · AnonReviewer4 · 2020-11-19
> > **Baselines**
> >
> > The baselines seem a little low, for example on CIFAR-100 with WRN28-10, the original paper reports an accuracy of 81.7% vs 77.0% of the authors. What is the reason of this mismatch? Without having a properly tuned baseline it is hard to tell if these automatic learning rates have performance close to the fine tuned one..

---

> > > ### Author Response · Authors · 2020-11-20
> > > **Response to Review #4 about baselines.**
> > >
> > > Thanks for your question.  The reporting accuracy of SGD+ MultiStep in Table 2 is 77.04,  since here the Momentum = 0.  We  have added the reporting accuracy of SGDM+ MultiStep in Table 3 with 'Momentum = 0.9' in the updated version, where the setting is same with the original paper.  Now the accuracy of  SGD+ MultiStep in Table 3 is 80.74 vs 80.75 of the original paper (with test error 19.25 in Table 3 of the original paper).
> > >
> > > As shown in Table 3, our MLR-SNet can help SGD (Momtume=0) get close to the best baselines (SGDM+MultiStep, Momentum = 0.9) by 79.41% vs 80.75. And under the same experimental setting, the MLR-SNet help SGD (Momtume=0)  outperform the baseline SGD+MultiStep by  79.41% vs 77.04 in Table 2.  Here we report accuracy of SGD (Momtume=0)  of MLR-SNet, since we treat image and text tasks in a unified learning framework with the same SGD algorithm setting (Momtume=0).

---

### Official Review · AnonReviewer2 · 2020-10-30
**Solid idea, but execution not up to the standard of ICLR**

**Rating:** 4
**Confidence:** 4

**Review:**

## Summary

This paper proposes a learning-to-learn type approach for step size schedules. An LSTM is used to predict step sizes based on observed values of the training loss. It is meta-trained with the goal of achieving maximal validation loss. The proposed method is evaluated empirically on image and text classification problems.


## Rating

The basic idea of the paper is solid. Fully meta-learned optimization methods have shown promising results, but have been brittle when transferred to settings that deviate substantially from those they have been meta-trained on. Using a standard optimizer update direction and only meta-learning the step size schedule seems like a reasonable idea to tackle the major pain point (step size tuning) while potentially providing more robustness. But, in my opinion, the idea is not executed and evaluated up to the standard of this conference. I have several issues with the quality and transparency of the experimental evaluation and the proposed method shows only limited success in the presented experiments. I am detailing my concerns below. Moreover, the paper is not really well-written. **In its current state, I recommend rejecting this paper.**


## Major Comments

1) The quality of the empirical evaluation is questionable in my opinion.
    a) The settings and competing methods are chosen somewhat erratically. For example, why is there no comparison to a simple multi-step schedule and/or SGDM for the text datasets?
    b) There is no explanation whatsoever of the used protocol for tuning the hyperparameters of the methods involved in the experimental comparison.
    c) Why is there no comparison to other automatic adaptation techniques (L4, HD) in the transfer experiments? The paper says that “Since the methods […] are not able to generalize, we do not compare them here”. I don’t understand what that is supposed to mean. Since they aren’t meta-learned, there is no notion of generalization for this method. But they could still be applied in the transfer setting as baselines.

2) This is a purely empirical paper proposing a practical method for learning rate scheduling. The impact of such a paper comes down, to a large extent, to the success of the proposed method. Unfortunately, the results are really a mixed bag. The proposed method outperforms baselines only in a small subset of the experiments: mainly when meta-training the MLR-SNet on the identical dataset/architecture (Section 4.1), and even there it gets outperformed by a multi-step schedule when using SGD with momentum. In the transfer experiments (Section 4.2), the proposed method is matched or outperformed by one of the simple baselines in almost all cases. Of course, there might be an argument to be made in terms of the effort of manual tuning that goes into the different baselines. But that point is not really driven home in the paper and I am skeptical for two reasons: (i) the absence of a clear hyperparameter tuning protocol (see previous point) and (ii) the fact that the method still relies on a manually-chosen scaling factor $\gamma$ for the step size. Overall, I don’t think the experimental results are convincing.

3) I am missing a discussion of the computational and memory cost of the proposed method. The meta-updates of the LSTM weights (line 6 in Algorithm 1) requires backpropagating through an interval of $T_\text{val}$ past iterations. Such back propagation through unrolled training trajectories are usually very memory-intensive (the entire state of the model has to be kept in memory for the unrolled iterates)  and adds considerable computational cost. In Section 4.1, this supposedly very costly method is compared to baselines that are essentially for free without any consideration given to that discrepancy in computational cost. Of course, this is a one-off cost when using a pre-trained MLR-SNet as a plug-and-play step size scheduler, but the results for this transfer setting are not really convincing (see above).

4) The paper proposes to adapt the learning rate based on the observed loss values. Fully meta-learned optimizers use gradient observations in addition to the loss. Why did you choose not to use the gradient information. To keep the method light-weight? Because a good step size can be chosen exclusively based on loss values? I think this modelling choice should be discussed and justified in the paper.


## Minor Comments

5) The quality of the writing is subpar. I am trying not to get hung up on linguistic mistakes, but occasionally it is really hard to decipher the meaning of certain sentences. There are also plenty of typos and stylistic errors. The bib-file could need a thorough review: you are citing arXiv versions of several papers that have been published in peer-reviewed venues, conferences are cited only by their acronym, et cetera.

6) It would have been nice to investigate the versatility of the proposed method by applying it to other base update directions than SGD(M). For example, could the MLR-SNet be applied on top of Adam as well?

---

> ### Author Response · Authors · 2020-11-18
> **Response to Reviewer #2 （Part 1）**
>
> Thank you for the constructive comments! We really appreciate the comments for improving the clarity of statements and experimental verifications.
>
> Q2.1 No multi-step schedule and/or SGDM for the text datasets.
>
> A.	As point out in [1,2], for the specific task of neural language modeling, traditionally SGD without momentum has been found to outperform other algorithms such as momentum SGD. The common strategy employed in language modeling is to reduce the learning rates by a fixed proportion when the performance of the model’s primary metric (such as perplexity) worsens or stagnates. Here, we adopt the strategy in the standard Pytorch library [3]. Actually, multi-step schedule performs worse than this strategy in language modeling. We adopt such strategy as compared methods for a strong baselines.
> In practice, it always needs to re-design a proper LR schedule for new tasks, such as different LR schedules for image and text tasks, and sometimes needs the additional expert knowledge.  To decrease the cost and increase the ability of scheduling LR for real applications,  we provide a flexible and explicit mapping formulation to parameterize LR, so that the proposed method can adaptively adjust LR according to training dynamics for different tasks in a unify data-driven framework.
>
> Q2.2 The hyperparameter tuning protocol.
>
> A. Thanks for your suggestion. We give detailed description for hyperparameters setting of the compared methods in Appendix B.
>
> Q2.3 No automatic adaptation techniques (L4, HD) in the transfer experiments.
>
> A. On the one hand, these methods have no explicit structure to guarantee that the learned LR schedules can be transferred to new learning tasks. When encountering new tasks, these methods need to re-learn the LR schedules without employing the learned knowledge from previous tasks. This process is time and computation expensive. However, our MLR-SNet can transfer learned knowledge through the explicit mapping formulation, which makes it plug-and-play to learn the new tasks. In the transfer experiments, unlike automatic adaptation techniques, hand-designed LR schedules have explicit formulation, thus they are treated as compared methods.
> One the other hand, in the section 4.1, it can be seen that hand-designed LR schedules perform much better than automatic adaptation techniques. Our MLR-SNet that has the similar performance with hand-designed LR schedules can also perform better than automatic adaptation techniques, without extra re-learning cost.
>
> Q2.4 About the theory of MLR-SNet, as well as the computational and memory cost.
>
> A.	Thanks for your suggestion. We provide the convergence analysis of the MLR-SNet, which is presented in Theorem 1 of the updated version [Section 5.1]. This theoretical result guarantees that our algorithm is convergent, and the empirical results verify this point. Also, we provide the computational complexity analysis of the MLR-SNet in both meta-training and meta-test stage [Section 5.3]. Our MLR-SNet takes barely longer time to complete the meta-training and meta-testing phase compared to hand-designed LR schedules. Therefore our method is completely capable of practical application.
>
> Q2.5 Why MLR-SNet performs similar with baselines.
>
> A.	Thanks for your questions, we will clarify in revision. Actually, the performance of the hand-design LR schedules can be regarded as the best/upper performance bound. Since these strategies have been tested to work well for the specific tasks, and they are written into the standard deep learning library. Our MLR-SNet can achieve the similar or even a little better performance compared with the best baselines for different tasks, which implies the effectiveness and generality of our method.
>
>
>
>
>
>
>
> ====================================================================================
>
> References
>
> [1] I. Sutskever, J. Martens, G. Dahl, and G. Hinton. On the importance of initialization and momentum in deep learning. In ICML 2013.
>
> [2] Merity S, Keskar N S, Socher R. Regularizing and optimizing LSTM language models. In ICLR, 2018.
>
> [3] https://github.com/pytorch/examples/tree/master/word_language_model.

---

> > ### Author Response · Authors · 2020-11-18
> > **Response to Reviewer #2 （Part 2）**
> >
> > Q2.6 Manually-chosen scaling factor $\gamma$.
> >
> > A. We set $\gamma=1$ for all image tasks, and $\gamma=40$ for all text tasks. Different $\gamma$ is due to the significant difference between image and text, and the existing hand-design and automatic adaptation methods need to carefully search the initial LR, which is very sensitive for the final performance. Also, the existing automatic adaptation methods only focus on the image tasks, and it is hard to obtain a satified initial LR for text tasks.  However, our strategy relaxes the requirements for tuning the exact initial LR, and allow the MLR-SNet to learn the LR from a bigger search space and decrease the hardness of tuning the exact initial LR.  In the Appendix B of the updated version [Fig12(c)], we study the influence on the DNN training performance of the different $\gamma$ values.  The ablation result shows that the MLR-SNet is robust to the choose of the $\gamma$ value. This implies that our MLR-SNet is potentially valid and robust for practical applications.
> >
> > Q2.7 Why not use gradient information.
> >
> > A.	Thanks for your suggestions. On the one hand, we observe that current hand-designed LR schedules often reduce the learning rates by a fixed proportion when the performance of the model’s loss worsens or stagnates. Recently, Rolinek et al., [4] proposed LR adaptation scheme based on the loss function. Therefore, we use the loss information to predict LR.  If we use the gradient as the input as the MLR-SNet, it is hard for MLR-SNet to generalize to the heavy-weight DNNs. While loss-based MLR-SNet can easily generalize to various problems, which satisfies the goal of our paper. We will clarify and discuss this in revision.
> >
> > Q2.8 Apply MLR-SNet on the top of Adam.
> >
> > A.	Thanks for your suggestions. We provide the additional experiments in the updated version [Fig. 16 in Appendix F ]. The preliminary experimental results shows our method can be applied on top of Adam. This implies that our methods is effective for different optimizer.
> >
> >
> >
> > =================================================================================
> >
> > References
> >
> > [4] Michal Rolinek and Georg Martius. L4: Practical loss-based stepsize adaptation for deep learning. In NeurIPS, 2018.

---

### Official Review · AnonReviewer1 · 2020-10-30
**Interesting work on important problem. Novelty and transferability need more justifications.**

**Rating:** 5
**Confidence:** 4

**Review:**

This work proposes to learn an adaptive LR schedule, which can adjust LR based on current
training loss and the information delivered from past training histories. It adopts a (parameterized) LSTM as the schedule, which is meta-learned following the existing work meta-weight-net. Extensive experiments are conducted to show its effectiveness on image classification, compared to pre-defined policies.

In general, this paper addresses an important problem of LR schedule and the suggested method is easy to follow and the experimental results are promising. However, there are several issues that need to be addressed to improve the work further.

1.	The novelty seems quite limited – it appears to be a straightforward application of meta-weight-net to the problem of LR scheduling.
2.	Analysis on the learned optimizers is lacking, such as the convergence and speed.
3.	It is not clear why the learned optimizer can be transferred to new heterogeneous tasks. Some deeper insights might be helpful. For example, can a LR scheduler learned from an image classification task be transferrable to a task of object detection?
4.	Other forms of parameterized LR scheduler should be discussed and compared, such as MLP. It is not obviously clear why LSTM is a good choice -- the structure of LSTM seems quite complicated.
5.	Missing relevant work: Meta-SGD (https://arxiv.org/abs/1707.09835) which also meta-learns a LR scheduler.

---

> ### Author Response · Authors · 2020-11-18
> **Response to Reviewer #1**
>
> Thank you for the constructive comments! We really appreciate the comments for improving the clarity of statements and experimental verifications.
>
> Q1.1 The novelty seems quite limited.
>
> A The MLR-SNet is different from Meta-Weight-Net on the following points:
> 1)	The learning object of the meta-learner. The Meta-Weight-Net learns the sample weights for loss function to handle the robust deep learning problems. The MLR-SNet schedules the learning rate for SGD algorithm to help the DNNs training. The former tries to revise the loss function of each samples, while the latter adaptively predictes the learning rate schedules for the whole training algorithm design.
> 2)	The structure of the meta-learner. The Meta-Weight-Net focus on the direct loss-weights mapping problem, thus the MLP can handle this. While MLR-SNet needs to predict the LR in the training process, which is long-term information dependent, and the LSTM is appropriate for dealing with such problem.
> 3)	The generalization of the meta-learner. The Meta-Weight-Net pays more attention to enhance the robustness of current robust learning tasks. While MLR-SNet focus more on the generalization of the meta-learner, and the learned MLR-SNet is plug-and-play and easily to transfer new heterogeneous tasks.
>
> Q1.2 Convergence and speed analysis of the learned optimizers.
>
> A.	Thanks for your suggestion. We provide the convergence analysis of the MLR-SNet, which is presented in Theorem 1 of the updated version [Section 5.1]. In Section 5.3, we provide the computational complexity analysis of the MLR-SNet in both meta-training and meta-test stage.
>
> Q1.3 Why the learned optimizer can be transferred to new heterogeneous tasks.
>
> A.	Thanks for your question. This is the main difference between our method and the automatic LR adaptation techniques. With an explicit parameterized structure, the meta-trained MLR-SNet can direct predict LR according to the input loss and training dynamics of the new tasks, and does not need to re-train the MLR-SNet. The pioneer work [1] also use the parameterized structure to learn the gradients, and then transfer the learned meta-learner to new tasks.  In a word, such parameterized structure with an explicit function mapping is capable of transferring the learned meta-learner to new heterogeneous tasks.
>
> Q1.4 About the structure of MLR-SNet.
>
> A.	Actually, the learning rate scheduling is a long-term information dependent problem, in which the current learning rate should be determined by the current training dynamics and the past training history. To parameterize the learning rate schedule with an explicit mapping formulation, the used meta-learner should have the capability to face such requirements. As we known, the LSTM is appropriate for such problem. The existing methods like [1-4] also use LSTM as meta-learner, but they use it to parameterize the gradients. However, MLP network can only deal with the direct function mapping problem, while ignore the temporal information in the training process.
> Furthermore, we conduct ablation experiments in the updated version [Section 5.2], and the experimental results show that LSTM meta-learner outperforms MLP meta-learner.
>
> Q1.5 Missing relevant work.
>
> A.	Thanks for providing this relevant work [5]. We’ll cite and discuss this paper in revision.
>
> ======================================================================================================
>
> References
>
> [1] Marcin Andrychowicz, Misha Denil, Sergio Gomez, Matthew W Hoffman, David Pfau, Tom Schaul, Brendan Shillingford, and Nando De Freitas. Learning to learn by gradient descent by gradient descent. In NeurIPS, 2016.
>
> [2] Sachin Ravi and Hugo Larochelle. Optimization as a model for few-shot learning. In ICLR, 2017.
>
> [3] Yutian Chen, Matthew W Hoffman, Sergio Gómez Colmenarejo, Misha Denil, Timothy P Lillicrap, Matt Botvinick, and Nando De Freitas. Learning to learn without gradient descent by gradient descent. In ICML, 2017.
>
> [4] OlgaWichrowska, Niru Maheswaranathan, MatthewWHoffman, Sergio Gomez Colmenarejo, Misha Denil, Nando de Freitas, and Jascha Sohl-Dickstein. Learned optimizers that scale and generalize. In ICML, 2017.
>
> [5] Li Z, Zhou F, Chen F, et al. Meta-sgd: Learning to learn quickly for few-shot learning. arXiv preprint arXiv:1707.09835, 2017.

---

### Author Response · Authors · 2020-11-19
**Summary of revisions**

We sincerely appreciate all reviewers for their valuable and constructive comments, which make our paper better without doubt.

We have meticulously addressed their mentioned problems and revised our manuscript based on their valuable suggestions. In particular, we have made the following key changes:

1) We have added the convergence analysis of the MLR-SNet in Theorem 1 of the Section 5.1 , as well as the computational and memory cost in Section 5.2 (pointed out by Reviewer #1 and Reviewer #2).

2) We have added the ablation experiments of the structure choose of the MLR-SNet in Section 5.2 (pointed out by Reviewer #1).

3) We have added the ablation experiments of the choose of the $\gamma$ values in Fig.12(c) in Appendix B (pointed out by Reviewer #2 and Reviewer #4).

4) We have provided the preliminary exploration of the influence on meta-test tasks of the different meta-training tasks in Appendix E (suggested by Reviewer #3).

5) We have studied that applying MLR-SNet on Top of Adam to demostrate the versatility of  the MLR-SNet in Appendix F (suggested by Reviewer #2).

6) We have revised one paragraph in Section 4.1 to show why our MLR-SNet bring a little performance increase compared with baselines (pointed out by Reviewer #2).

7) We have reported the accuracies in Table 2-5 in appendix (suggested by Reviewer #4).

---

### Decision · Program_Chairs · 2021-01-07
**Final Decision**

**Decision:**

Reject

**Comment:**

The paper proposed a meta-learning method for tuning the learning rate. In the discussion, reviewers agreed that the key issue is that the empirical evaluation is not yet sufficient to demonstrate the efficacy of the method. In particular, this is an especially pressing issue given that there are now many meta-learning methods for tuning the learning rate (none popular in practice though), and the paper does not compare to any of them. Relatedly, most reviewers found that the novelty of the method is not clearly established and discussed in the paper.

Based on the above, I have to recommend rejecting the paper. I would like to thank the authors for submitting the work for consideration to ICLR. I hope the feedback will be useful for improving the work.